# USP11 controls R-loops by regulating senataxin proteostasis

Mateusz Jurga[1,2], Arwa A. Abugable[1], Alastair S. H. Goldman[3] & Sherif F. El-Khamisy [1,2✉]

R-loops are by-products of transcription that must be tightly regulated to maintain genomic stability and gene expression. Here, we describe a mechanism for the regulation of the R-loop-specific helicase, senataxin (SETX), and identify the ubiquitin specific peptidase 11 (USP11) as an R-loop regulator. USP11 de-ubiquitinates SETX and its depletion increases SETX K48-ubiquitination and protein turnover. Loss of USP11 decreases SETX steady-state levels and reduces R-loop dissolution. Ageing of USP11 knockout cells restores SETX levels via compensatory transcriptional downregulation of the E3 ubiquitin ligase, KEAP1. Loss of USP11 reduces SETX enrichment at *KEAP1* promoter, leading to R-loop accumulation, enrichment of the endonuclease XPF and formation of double-strand breaks. Overexpression of KEAP1 increases SETX K48-ubiquitination, promotes its degradation and R-loop accumulation. These data define a ubiquitination-dependent mechanism for SETX regulation, which is controlled by the opposing activities of USP11 and KEAP1 with broad applications for cancer and neurological disease.

[1] School of Bioscience, Department of Molecular Biology and Biotechnology, The Healthy Lifespan Institute and the Institute of Neuroscience, University of Sheffield, Sheffield, UK. [2] The Institute of Cancer Therapeutics, University of Bradford, Bradford, UK. [3] Faculty of Life Sciences, University of Bradford, Bradford, UK. ✉email: s.el-khamisy@sheffield.ac.uk

DNA:RNA hybrids (R-loops) are natural byproducts of transcription. Structural studies suggest a 'thread back model' of R-loop formation, whereby a nascent RNA invades the DNA duplex and anneals to the template DNA strand, causing the non-template DNA strand to be left un-annealed[1]. R-loop formation is promoted over G-rich regions and at sites of RNA polymerase stalling[2,3]. R-loops occupy ~5% of human and mouse genomes, and are usually found over promoter and termination regions[4]. As R-loops are more thermo-dynamically stable than double-stranded DNA, they persist until enzymatically resolved[5]. There are positive and negative consequences of R-loops. For example, R-loops facilitate class switch recombination, transcription-associated homologous recombination and transcription termination[6–8], regulate gene expression[9] and DNA methylation[10]. On the other hand, aberrant accumulation of R-loops causes genome instability[11–17]. Thus, a tight control of R-loop homeostasis is essential to promote their beneficial effects and avoid their deleterious consequences.

Human cells possess R-loop-specific nucleases such as RNase H1, RNase H2 and R-loop-specific helicases, such as DHX9, DDX19, DDX21 and senataxin (SETX)[18–20]. RNA-binding proteins were also shown to prevent R-loop formation[11,21]. In human mitochondria, mtEXO degradosome, which contains a ribonuclease and a helicase, controls R-loop levels[22]. SETX was shown to resolve R-loops at transcription termination sites of RNA Polymerase II (RNAPII)[8] and more recently at sites of double-strand breaks[23]. The importance of SETX function is highlighted by non-sense loss-of-function mutations of *SETX*, which cause ataxia with oculomotor apraxia type 2, and missense gain-of-function mutations, which cause a juvenile form of amyotrophic lateral sclerosis type 4 (ALS4)[24,25]. The ALS4 SETX$^{L389S}$ variant was shown to resolve R-loops at 1200 human promoters leading to transcriptional misregulation[26]. Furthermore, we recently reported a role for SETX in cancer, by protecting cells from R-loop-mediated DNA damage in hypoxia, which is a major cause of resistance to anti-cancer therapeutics[27].

Although much is known about R-loop metabolizing enzymes, little is known about the mechanisms that regulate their homeostasis. Protein homeostasis is controlled by many processes such as ubiquitination and deubiquitination. Ubiquitination is a process of conjugating small ubiquitin peptides (8.5 kDa) to lysine residues of target proteins. As ubiquitin itself possesses lysine residues, ubiquitin chains and branches can be formed by E3 ligases and trimmed by deubiquitinases (DUBs). Ubiquitination has been shown to signal for proteasomal degradation, influence subcellular localization and alter protein–protein interactions or enzymatic activities[28]. DUBs have been shown to promote DNA damage signalling and repair mechanisms. For example, USP1 has been suggested to inhibit translesion synthesis[29]. USP47 regulates the steady-state levels of DNA polymerase-β, which is essential for base excision repair[30]. USP3 and USP16 control cell cycle progression via deubiquitination of histone 2A (H2A)[31,32]. UCHL3 fine tunes topoisomerase-linked DNA break repair via deubiquitination of TDP1[33,34]. The activity of p53 is also regulated by DUBs[35], which further highlights the role of deubiquitination in the maintenance of genome integrity.

Here we conducted an unbiased genetic screen coupled with cellular and biochemical analyses to reveal the ubiquitin-specific peptidase 11 (USP11) as an R-loop regulator via controlling SETX proteostasis. USP11-deficient cells exhibit high levels of R-loops, which are reduced by expression of wild-type but not catalytically inactive USP11. Depletion of USP11 or SETX alone increases detectable R-loops and causes hypersensitivity to topoisomerase I poisons, and depletion of both does not exacerbate the R-loop and hypersensitivity phenotypes, indicating that USP11 and SETX are in the same pathway. Consistent with this, SETX interacts with USP11 and the interaction is increased by R-loop induction in a transcription-dependent manner. Depletion of USP11 increases SETX K48-ubiquitination and degradation, which leads to accumulation of R-loops that are converted to double-strand breaks. Finally, we describe a mechanism of cellular adaptation to USP11 loss through transcriptional down-regulation of the E3 ligase, KEAP1, which counteracts USP11 to maintain SETX protein levels.

## Results

**Proteasomal inhibition perturbs R-loop homeostasis.** R-loop homeostasis is controlled by helicases and ribonucleases; however, little is known about how R-loop-metabolizing enzymes are regulated. To examine this, we first set out to validate the specificity of the widely employed R-loop detection method using S9.6 antibodies in MRC-5 cells. Cells were incubated with the topoisomerase inhibitor camptothecin (CPT) for 10 min and S9.6 signal was examined by immunofluorescence (Fig. 1a). The co-localization of S9.6 with α-nucleolin suggests nucleolar enrichment of R-loops (Fig. 1a), as CPT treatment did not change the number of nucleolin foci (Supplementary Fig. 1a). The number and intensity of nucleolar S9.6 foci increased after 10 min CPT treatment (Fig. 1b), which is consistent with previous findings[3,36]. By masking the nucleolar S9.6 foci, we observed ~40% increase in nuclear S9.6 signal intensity (Fig. 1b). The CPT-dependent increase of S9.6 signal was specific to R-loops, as it was ablated by pre-treatment with the R-loop nuclease *Escherichia coli* RNase-H (ec-RH) (Fig. 1c). Furthermore, depletion of the R-loop helicase, SETX, increased nucleolar S9.6 signal, which was also reversed by ec-RH treatment (Fig. 1d).

To test whether perturbed protein degradation impacts R-loop homeostasis, MRC-5 and HEK-293 cells were treated with the proteasomal inhibitor MG132 and examined for R-loop levels. MG132 treatment increased the nucleolar S9.6 foci measured by immunofluorescence (Fig. 1e) and by DNA/RNA immunoprecipitation (DRIP) followed by quantitative PCR (qPCR) (Fig. 1f and Supplementary Fig. 1b). We noted that DRIP uncovered an MG132-dependent increase in R-loops at both RNAPI and RNAPII loci such as *28S* and *ING3*, respectively. These data suggest that physiological levels of proteasomal degradation are required for R-loop homeostasis.

**DUB siRNA screen identifies USP11 as a novel R-loop regulator.** We next reasoned that an unbalanced ubiquitin regulation could perturb R-loop homeostasis. The covalently conjugated ubiquitin is reversed by DUBs[37]. If a DUB facilitates deubiquitination of a protein that is involved in R-loop regulation, then a knockdown of this DUB should lead to proteasomal degradation of the target protein and subsequently alter R-loop levels. To test this, we carried out a high-throughput DUB screen using S9.6 immunostaining as a readout. We individually depleted 99 human DUBs in MRC-5 cells using small interfering RNA (siRNA) pools in two different conditions: unchallenged and CPT-treated (Fig. 1g). The S9.6 immunostaining signal was quantified using MetaXpress® software and the screen was repeated three times. The primary screen uncovered nine candidates whose knockdown led to either an increase or decrease in R-loops. The raw data are deposited in Supplementary Data 1. Depletion of eight of the identified DUBs triggered an aberrant R-loop levels in either untreated or CPT-treated cells. The depletion of only one DUB (USP11) consistently caused R-loop accumulation in both tested conditions (untreated and CPT, Fig. 1h), suggesting that USP11 is likely a major regulator of R-loops dissolution.

Deconvolution of the siRNA pool used to knock down each candidate followed by depletion of USP11 and subsequent R-loop

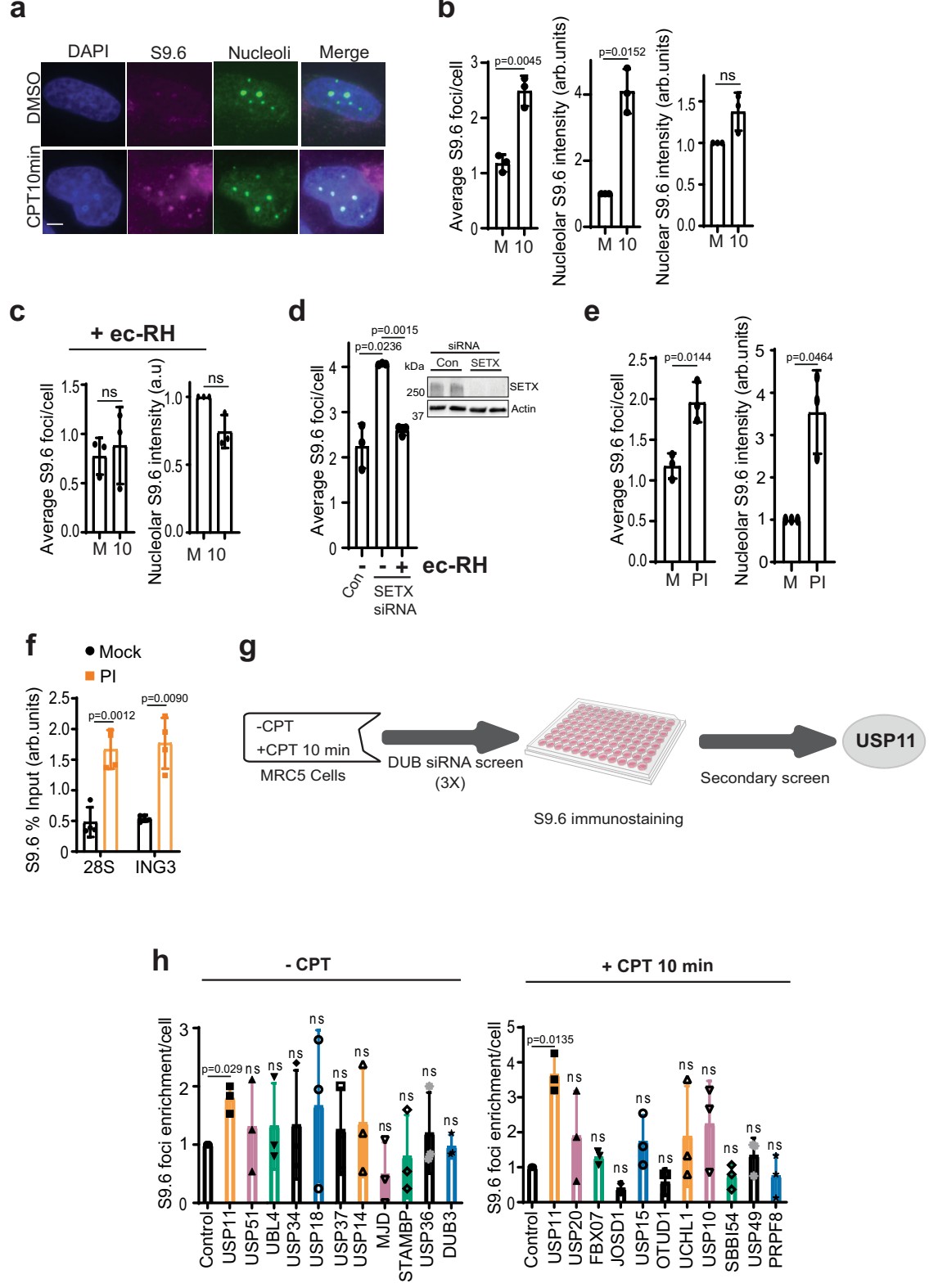

accumulation was validated using four siRNA sequences separately and together (Fig. 2a). The increase in nucleolar S9.6 signal upon USP11 depletion was specific to R-loops, as it disappeared in cells ectopically expressing green fluorescent protein (GFP)-RNase H1 (Fig. 2b). Further, this increase was not specific to MRC-5 cells, as it was also observed in U2-OS cells (Supplementary Fig. 1c). Notably, depletion of USP11 led to an increase in detectable R-loops in both the nucleolar (*28S, R7*) and

nuclear (*ING3, Actin*) loci, as measured by DRIP (Fig. 2c and Supplementary Fig. 1d). We note that under our experimental conditions, S9.6 immunofluorescence did not reveal a statistically significant increase of nuclear R-loop signal upon transient USP11 depletion, which is likely due to the semi-quantitative nature of the immunostaining and imaging as compared to quantitative DRIP-qPCR. We therefore validated nucleolar and nuclear R-loop accumulation in USP11-depleted cells by

**Fig. 1 A genetic screen uncovers USP11 as a novel R-loop regulator. a** MRC-5 cells were treated with DMSO or 25 μM camptothecin (CPT) for 10 min and immediately harvested for S9.6/nucleolin immunofluorescence. Representative confocal images from three biological repeats are shown, scale bar = 3 μm. **b** MRC-5 cells were treated with DMSO (Mock; M) or 25 μM CPT (10) for 10 min and immediately collected for S9.6/nucleolin immunofluorescence. Data are the average ± SD from 3 biological repeats, each containing at least 100 cells. The average number of S9.6 foci/cell was calculated (left panel). The total nucleolar (middle panel) and nuclear (right panel) S9.6 fluorescence was measured using ImageJ and normalized to mock. ns; $p > 0.05$, two-tailed Student's $t$-test. **c** MRC-5 cells were treated with DMSO (Mock; M) or 25 μM CPT (10) and then incubated with RNase-H (ec-RH). Cells were then processed for S9.6/nucleolin immunostaining. Data are the average ± SD from 3 biological repeats, each containing at least 100 cells. The average number of S9.6 foci/cell was calculated (left panel) and total nucleolar S9.6 fluorescence was measured and normalized to mock (right panel). ns; $p > 0.05$, two-tailed Student's $t$-test. **d** MRC-5 cells were transfected with SETX or scrambled (Con) siRNAs and incubated with RNase-H (ec-RH). Cells were processed for S9.6 immunostaining. The average number of S9.6 foci/cell was calculated and data represent average ± SD from three biological repeats, each containing at least 100 cells. Two-tailed Student's $t$-test. Insert, western blotting showing SETX protein expression with Actin as a loading control. **e** MRC-5 cells were incubated with DMSO (Mock; M) or 25 μM MG132 (proteasomal inhibitor, PI) followed by S9.6/nucleolin immunostaining. Data are the average ± SD from 3 biological repeats, each containing at least 100 cells. The average number of S9.6 foci/cell was calculated (left panel) and total nucleolar S9.6 fluorescence normalized to mock (right panel). *$p < 0.05$, two-tailed Student's $t$-test. **f** HEK-293 cells were mock incubated with DMSO or 25 μM MG132 (PI) followed by DNA/RNA immunoprecipitation (DRIP) using S9.6 antibodies. Quantitative PCR was conducted using primers targeting nucleolar (28 S) and nuclear (ING3) loci. The data represent the average ± SD from four biological repeats. Raw % input values are shown in Supplementary Fig. 1b. Two-tailed Student's $t$-test. **g** Flowchart depicting the design of S9.6 genetic screen. A secondary S9.6 screen was conducted in a 24-well format, which uncovered USP11 as a new R-loop regulator. **h** The DUB siRNA screen was performed as described above and data from selected DUBs are presented showing average S9.6 foci/cell normalized to scrambled siRNA controls from three biological repeats ± SD. ns; $p > 0.05$, two-tailed Student's $t$-test.

repeating the experiments using recently published DRIP-qPCR protocols[38], employing seven well-characterized primer pairs including two negative controls (SNRPN-neg and MYADM-neg) (Supplementary Fig. 1e, f). Together, these data suggest that USP11 regulates R-loop homeostasis at nucleolar and nuclear loci.

**The catalytic activity of USP11 is required to suppress perturbed R-loop levels.** Next, we examined whether a persistent knockout of USP11 also perturbs R-loop levels. To test this, we generated two USP11-knockout HEK-293 cell lines using CRISPR-Cas9. USP11-sgRNA clone 1 had a 23nt deletion and USP11-sgRNA clone-2 had a 2nt deletion, both in exon 1 of the USP11 gene (Supplementary Fig. 2a). As predicted from the sequence, both clones had no detectable levels of USP11 (Fig. 2d). We then complemented USP11-sgRNA clones 1 and 2 with wild-type USP11 or a catalytically inactive USP11[C318S] mutant[39–42] (Fig. 2d). In agreement with transient depletion using siRNA, deletion of USP11 led to elevated nucleolar R-loops as measured by S9.6 immunofluorescence (Fig. 2e, f). Complementation with wild-type USP11, but not the catalytically inactive USP11[C318S] mutant, reversed the perturbed accumulation of R-loops. This observation suggests an enzymatic role of USP11 to regulate R-loops. We noted that S9.6 immunostaining did not measure a detectable difference between control and USP11-knockout cells in the nucleoplasm (Supplementary Fig. 2b). However, deletion of USP11 led to increased R-loops at both nucleolar and nuclear loci when measured by DRIP (Fig. 2g and Supplementary Fig. 2c). The increase was specific to USP11 enzymatic activity, as complementation with wild-type USP11, but not USP11[C318S], reduced R-loop levels. This was further confirmed by a third orthogonal method using slot blot (Supplementary Fig. 2d, e). We conclude from these experiments that the catalytic activity of USP11 is required for the maintenance of physiological 'steady-state' levels of R-loops.

As loss of USP11 has been linked to genome instability[39,41,43], we reasoned that a proportion of DNA damage observed in USP11-deficient cells could be due to aberrant accumulation of R-loops. To test this, we performed an alkaline comet assay to measure chromosomal breaks in USP11-deficient cells with and without ectopic expression of GFP-RNase-H1. USP11-deficient cells possessed higher levels of DNA breaks than control cells, which were reduced by RNase H1 overexpression (Fig. 2h). Furthermore, depletion of USP11 led to hypersensitivity to CPT and Olaparib, which is consistent with previous reports[41], and the

hypersensitivity was rescued by overexpression of RNase H1 (Fig. 2i). Together, we conclude from these experiments that USP11 maintains genome integrity by regulating R-loop levels.

**USP11 and SETX act in the same pathway to regulate R-loop homeostasis.** We next set out to address how USP11 regulates R-loops, primarily guided by its effects on nucleolar R-loops. The N-terminal domain of SETX[1–667] has been shown to localize to the nucleoli in two independent studies[44,45]. SETX has also been reported in a complex with nucleolin, a key component of nucleoli[46]. Moreover, SETX[1–650] has been suggested to be ubiquitinated as shown by a band shift following treatment with N-Ethylmaleimide (NEM), a cysteine protease that inhibits most DUBs including USP11[47]. Our immunostaining is consistent with the above reports, showing nucleolar localization of SETX[1–667] (Supplementary Fig. 3a). Further, USP11 immunostaining showed both nuclear and nucleolar localization (Supplementary Fig. 3b). To test whether USP11 and SETX are epistatic, USP11 and SETX were depleted separately and together (Supplementary Fig. 3c). Interestingly, depletion of USP11 alone led to a mild, albeit significant, reduction of SETX protein levels, suggesting that USP11 might control SETX proteostasis. This effect was specific to SETX, because it did not impact the protein level of another enzyme involved in protein-linked DNA repair, TDP1 (Supplementary Fig. 3c). Depletion of SETX, USP11 alone or their co-depletion led to a similar accumulation of R-loops (Fig. 3a). Furthermore, co-depletion of USP11 and SETX led to the same hypersensitivity to R-loop-inducing agents such as CPT and formaldehyde (Fig. 3b and Supplementary Fig. 3d). To examine whether endogenous USP11 and SETX interact, we performed proximity ligation assays (PLAs). Under unperturbed conditions, we detected a PLA signal between USP11 and SETX, indicating their cellular proximity (Fig. 3c). The PLA signal was specific, as it was not detected when one of the antibodies was omitted (Supplementary Fig. 3e). Notably, inducing R-loops by CPT treatment led to ~5-fold increase in PLA signal. Furthermore, the interaction was transcription dependent, because pre-treatment with the transcription inhibitor, α-amanitin (AMN), suppressed the PLA signal (Fig. 3c). We next characterized the interaction between USP11 and SETX. The residues reported to mediate USP11 binding to its substrates[48] were also required for binding to SETX, as shown by the lack of PLA signal between endogenous SETX and the USP11-binding site mutant, GFP-USP11[L208F,S242R] (Supplementary Fig. 4a). Consistently,

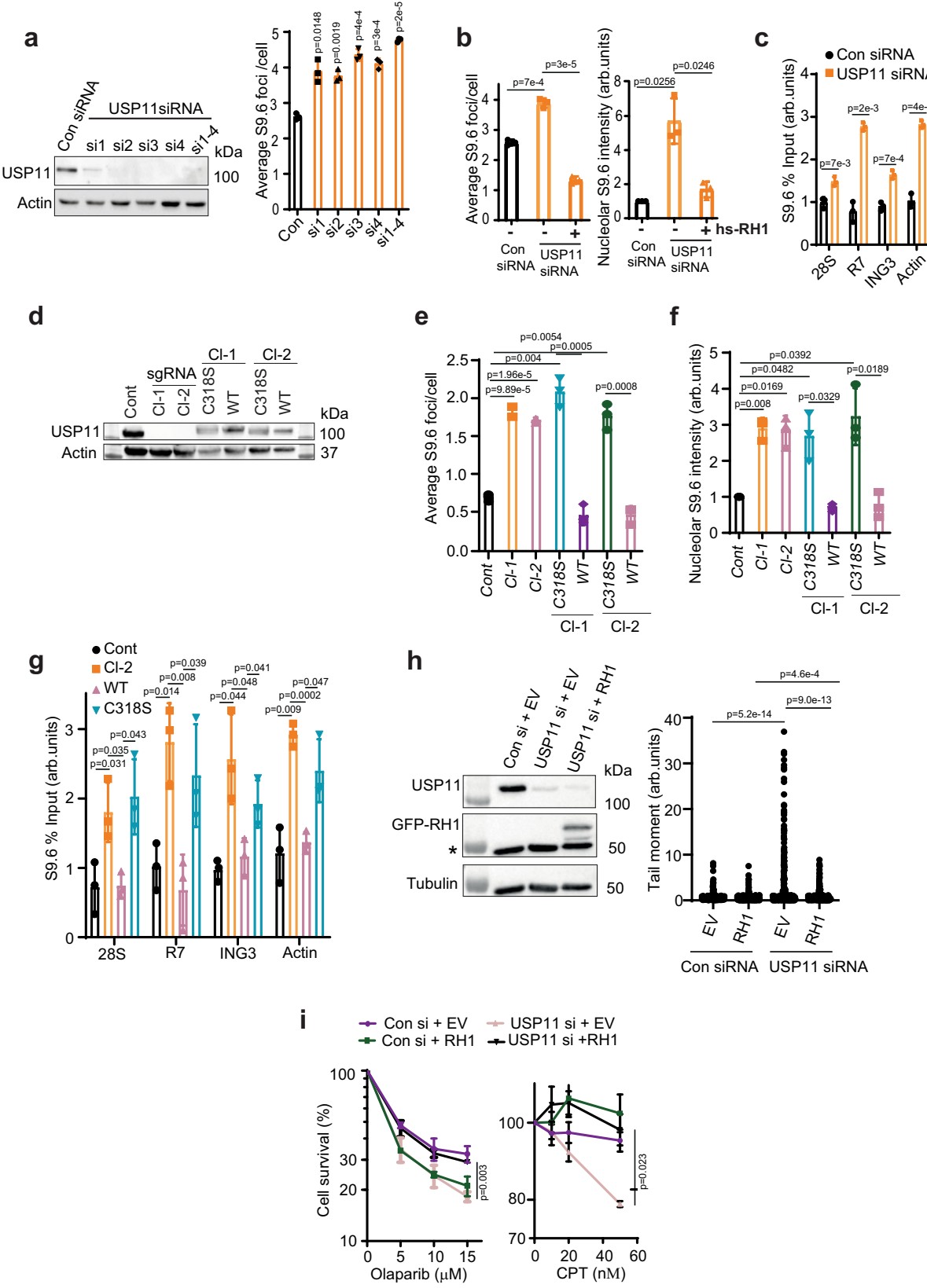

overexpression of wild-type USP11, but not USP11[L208F,S242R], in USP11-knockout cells reduced R-loop levels (Supplementary Fig. 4b). Finally, both USP11 and SETX were enriched in cells released to S phase following double-thymidine block (Fig. 3d, e),

which is consistent with *Saccharomyces cerevisiae* data showing the SETX orthologue Sen1p is cell cycle-regulated[49] and human data showing the same for USP11[41]. This provides evidence for a cell cycle-dependent regulation of SETX protein levels in human

**Fig. 2 Loss of USP11 triggers R-loop accumulation and R-loop-dependent DNA damage. a** MRC-5 cells were transfected with scrambled siRNA (Con), four different siRNA sequences targeting different regions in *USP11* (si1–si4) or pooled siRNA containing all four siRNAs (si1–4). Depletion of USP11 was examined by immunoblotting (left panel) and by immunofluorescence (right panel). The average number ± SD of S9.6 foci per cell was calculated from 3 biological repeats each containing at least 100 cells. **b** MRC-5 cells were transfected with USP11 or scrambled 50 nM siRNA (Con) and with human RNase H1 (hs-RH1). Cells were then processed for S9.6/nucleolin immunostaining. Data are the average ± SD from 3 biological repeats, each containing at least 100 cells and presented as the average number of S9.6 foci/cell (left panel) and mean S9.6 nucleolar intensity (right panel). *p*-Values calculated using two-tailed Student's *t*-test. **c** Lysates from Control (Con siRNA) and USP11-depleted cells (USP11 siRNA) were subjected to DNA/RNA immunoprecipitation (DRIP) using S9.6 antibodies. Quantitative PCR was conducted using primers targeting nucleolar (*28S* and *R7*) and nuclear (*ING3* and *actin*) loci. The data represent the average ± SD from three biological repeats. Raw % input values are shown in Supplementary Fig. 1d. *p*-Values calculated using two-tailed Student's *t*-test. **d** HEK-293 cells were transfected with a vector expressing Cas9 and sgRNA targeting exon 1 of USP11. USP11-knockout clones were then stably complemented by vectors encoding full-length FLAG-HA-USP11 (WT) or catalytically inactive FLAG-HA-USP11$^{C318S}$ mutant (C318S). The images are representative of three biological repeats. **e**, **f** USP11 sgRNA clones (Cl-1 and Cl-2) and those complemented with WT USP11 or the C318S mutant were examined by S9.6/nucleolin immunofluorescence. The average number of S9.6 foci/cell (**e**) and total nucleolar fluorescence normalized to control (Cont) cells (**f**) were calculated from 3 biological repeats, each containing at least 100 cells and presented as average ± SD. *p*-Values calculated using two-tailed Student's *t*-test. **g** Lysates from Control (Cont) and USP11 sgRNA Cl-2 and USP11 sgRNA Cl-2 complemented with USP11 (WT) or catalytically inactive USP11$^{C318S}$ mutant (C318S) were subjected to DNA/RNA immunoprecipitation using S9.6 antibodies. Quantitative PCR was conducted using primers targeting nucleolar (*28S* and *R7*) and nuclear (*ING3* and *actin*) loci. The data represent the average ± SD from three biological repeats. Raw % input values are shown in Supplementary Fig. 2c. *p*-Values calculated using two-tailed Student's *t*-test. **h** Control (Con si) and USP11-depleted MRC-5 cells (USP11si) were transfected with plasmids encoding eGFP-EV (EV) or eGFP-RNase H1 (RH1). The expression level of USP11, eGFP-Rnase H1 and Tubulin was analysed by immunoblotting. * denotes a nonspecific band (left panel). Chromosomal breaks were quantified by an alkaline comet assay and data represent the average of 3 biological replicates ± SD, each containing 125 cells (right panel). *p*-Values calculated using two-tailed Student's *t*-test. **i** MRC-5 cells transfected with Control (control) or USP11 siRNA and eGFP-EV (EV) or eGFP-RNase H1 (RH1) were incubated with the indicated doses of Olaparib or CPT and left to grow for 7 days. The surviving colonies were counted and % survival calculated relative to mock-treated cells. Data are the average ± SEM from three biological repeats. *p*-Values calculated using two-tailed Student's *t*-test.

cells. Together, these data show that USP11 and SETX interact physically and functionally to regulate R-loops in a cell cycle-dependent manner and suggest that SETX might be a substrate for USP11 DUB activity.

**USP11 controls SETX proteostasis.** The N-terminal domain of SETX$^{1–650}$ has been shown to be ubiquitinated in human cells[47], yet the DUB and the E3 ligase remain unknown. The *S. cerevisiae* orthologue of SETX, Sen1p, has also been shown to be ubiquitinated[49]. We reasoned that depletion of USP11 could promote SETX ubiquitination, which would then trigger its degradation, potentially explaining the elevated R-loop phenotype. In line with this hypothesis, both USP11-sgRNA clones displayed reduced SETX protein levels compared to controls (Fig. 3f). Complementation of USP11-sgRNA clones with wild-type USP11 increased endogenous SETX protein levels, whereas complementation was not possible with the catalytically inactive USP11$^{C318S}$ (Fig. 3g and Supplementary Fig. 4c). This effect was likely specific to SETX, because it did not impact the protein level of another enzyme involved in protein-linked DNA repair, spartan. The impact of the loss of USP11 on SETX protein levels was not due to changes in SETX mRNA levels, as shown by reverse-transcription qPCR (RT-qPCR, Fig. 3h). Furthermore, the protein turnover of SETX, but not TDP1 as control, was faster in USP11 sgRNA-knockout cells pre-incubated with the protein synthesis inhibitor, cycloheximide (CHX), compared to controls (Fig. 3i, j). Together, these data show that SETX proteostasis is regulated by USP11 and substantiate the proposition that SETX is a substrate for USP11 DUB activity.

**USP11 depletion increases SETX ubiquitination.** To test whether USP11 regulates SETX by controlling its ubiquitination, we examined the ubiquitination level of GFP-SETX$^{1–667}$, full-length FLAG-SETX and endogenous full-length SETX in control and USP11-depleted cells using pull-down assays under denaturing conditions. Higher levels of ubiquitinated SETX$^{1–667}$, ubiquitinated full-length FLAG-SETX and ubiquitinated endogenous full-length SETX following normalization to His-ubiquitin were observed in USP11-depleted cells, as compared to controls

(Fig. 4a, b and Supplementary Fig. 4d, e). This was further confirmed by incubating purified ubiquitinated GFP-SETX$^{1–667}$ with recombinant USP11. During the in vitro deubiquitination reaction, recombinant USP11 liberated SETX$^{1–667}$ from the SETX$^{1–667}$/ubiquitin-His-nickel beads, resulting in a reduction of the remaining bead-bound ubiquitinated SETX$^{1–667}$ (Fig. 4c, d). These data provide evidence that USP11 depletion leads to accumulation of ubiquitinated species of SETX$^{1–667}$, as both pulldowns were conducted under denaturing conditions. Furthermore, USP11knockout sgRNA Cl-2 cells complemented with wild-type USP11-possessed lower levels of ubiquitinated GFP-SETX$^{1–667}$ compared to cells expressing the catalytically inactive USP11$^{C318S}$ (Fig. 4e, f). We next attempted to determine the ubiquitin linkage on SETX by assessing K48-ubiquitination using denaturing pulldowns. The more GFP-SETX$^{1–667}$ pulled down, the more K48-ubiquitination was observed (Fig. 4g). We did not observe K48 signal in the empty vector-GFP negative control, suggesting that GFP-SETX$^{1–667}$ was modified with K48-ubiquitin chains (Fig. 4g). Next, we compared the K48-ubiquitination level of GFP-SETX$^{1–667}$ in control and USP11-deficient cells under denaturing conditions (Fig. 4h). Higher levels of K48-ubiquitinated GFP-SETX$^{1–667}$ following normalization to the bait (GFP-SETX$^{1–667}$) were observed in USP11-depleted cells compared to controls (Fig. 4i), indicating that K48-ubiquitinated SETX is a substrate for USP11. Finally, and consistent with a role for SETX during transcription, we observed that GFP-SETX$^{1–667}$ interacts with endogenous USP11, RNAPII and the largest subunit of RNAPI (RPA-194) (Fig. 4j). A SETX/RNAPI interaction in human cells is consistent with data from *S. cerevisiae* showing interaction of the SETX homologue, Sen1p, with yeast RNAPI[50].

**Aged USP11-knockout cells restore SETX and R-loop levels.** To gain further insight on the consequences of persistent loss of USP11, we monitored R-loops in USP11-sgRNA-knockout cells over 12 passages. USP11-sgRNA clones restored R-loops to levels comparable to control cells in a passage-dependent manner, as measured by S9.6 immunostaining (Fig. 5a and Supplementary Fig. 5a) and by DRIP (Fig. 5b and Supplementary Fig. 5b). The continued passage also led to restoration of SETX protein levels (Fig. 5c) and increased

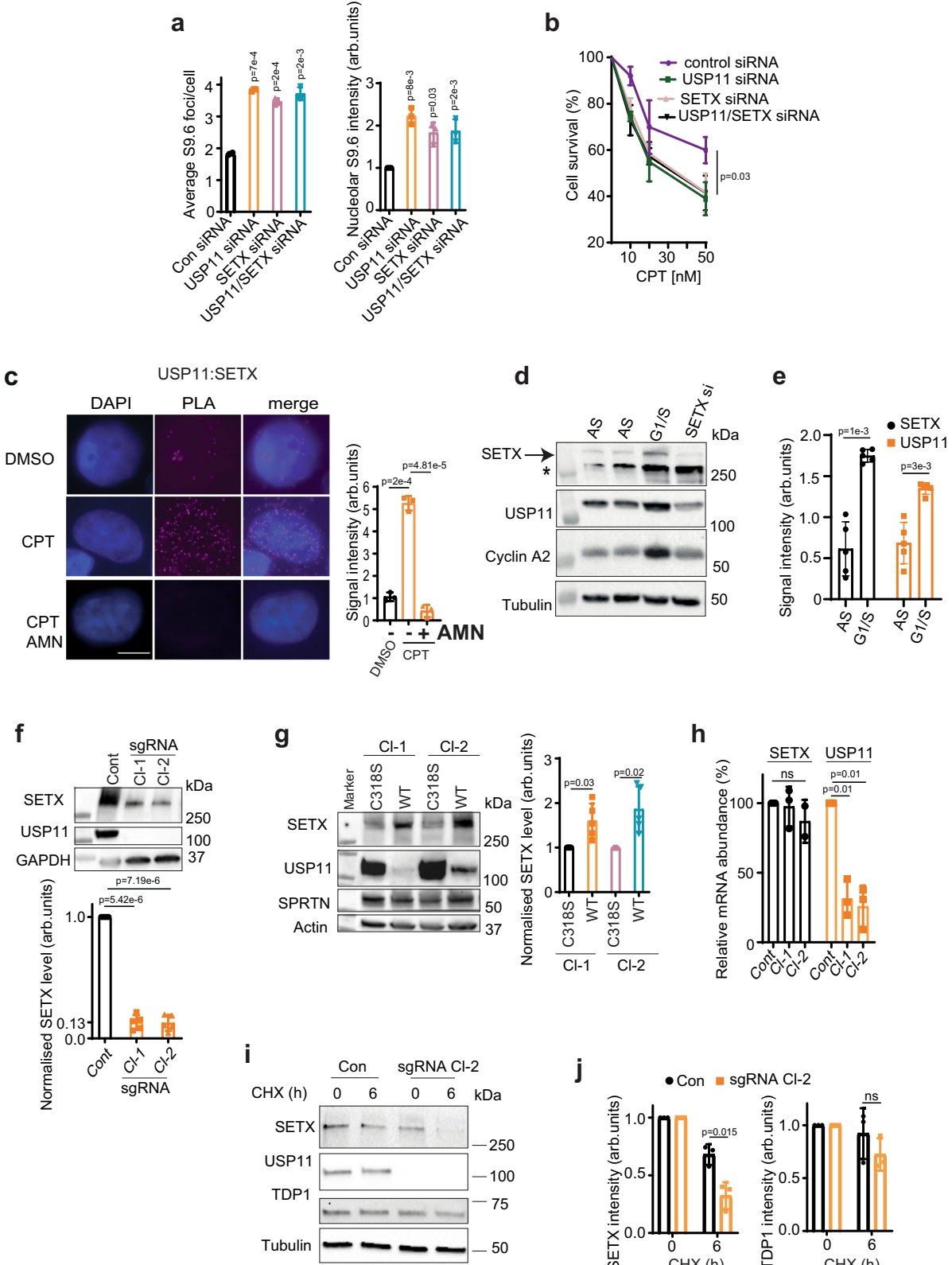

expression of the ageing senescence marker, p21 (Supplementary Fig. 5c). These findings were not due to restoration of USP11 levels, which remained undetectable in late passages (Fig. 5c). Furthermore, they were not due to age-related changes in SETX mRNA levels, which remained similar throughout the lifespan of both USP11-sgRNA clones (Fig. 5d). We therefore reasoned that the restoration of SETX protein levels was an adaptation to the continued loss of

USP11 via changes in SETX post-translational control rather than altered transcription rates. USP11 has been shown to promote the formation of BRCA1-PALB2-BRCA2 complex, which is opposed by the E3 ubiquitin ligase, KEAP1[41]. Interestingly, KEAP1 mRNA and protein levels were downregulated in the aged USP11-sgRNA clones (passage ≥ 12) when compared to early passages (Fig. 5e, f). Furthermore, overexpression of KEAP1 in aged USP11-knockout cells

**Fig. 3 USP11 and SETX work together to regulate R-loop homeostasis. a** MRC-5 cells treated with control (Con), USP11, SETX or USP11 and SETX siRNA were analysed by S9.6/nucleolin immunostaining. Data are the average ± SD from 3 biological repeats, each containing at least 100 cells and presented as average number of S9.6 foci/cell (left panel) and mean S9.6 nucleolar intensity (right panel). p-Values calculated using two-tailed Student's t-test. **b** MRC-5 cells treated with control (Con), USP11, SETX or USP11 and SETX siRNA were incubated with the indicated doses of CPT for 1 h and left to grow for 7 days. The surviving colonies were counted and % survival calculated relative to mock-treated cells. Data are the mean ± SD from three biological repeats. p-Values calculated using two-tailed Student's t-test. **c** MRC-5 cells mock treated with DMSO (DMSO), 25 μM CPT for 10 min or α-amanitin (AMN) overnight followed by CPT were subjected to proximity ligation assay using antibodies against endogenous USP11 and SETX. Representative images are shown, scale bar is equal to 10 μm (left panel). The signal intensity was measured using ImageJ (right panel). Data are the mean ± SD from three biological repeats. p-Values calculated using two-tailed Student's t-test. **d** HEK-293 cells were synchronized to G1/S boundary (G1/S) by a double-thymidine block. Asynchronized (AS) cells served as a control. The expression level of SETX, USP11, cyclin A2 and tubulin was analysed by immunoblotting. * denotes a nonspecific band. The images are representative of three biological repeats. **e** The band intensity of SETX and USP11 was quantified from five biological repeats of **d**, normalized to tubulin and then presented as fold increase of the mean ± SD compared to asynchronized samples. p-Values calculated using two-tailed Student's t-test. **f** Lysates from Control (Cont) and USP11-knockout HEK-293 cells (sgRNA Cl-1 and Cl-2) were fractionated by SDS-PAGE and analysed by immunoblotting using SETX, USP11 and GAPDH antibodies (top panel). SETX band intensities were normalized to GAPDH and presented as fold reduction compared to levels in control parental cells (bottom panel). Data are the average of five biological repeats and presented as mean ± SD. p-Values calculated using two-tailed Student's t-test two-tailed Student's t-test. **g** Lysates from USP11 sgRNA Cl-1 and Cl-2 complemented with wild-type USP11 (WT) or catalytically inactive USP11$^{C318S}$ mutant (C318S) were fractionated by SDS-PAGE and analysed by immunoblotting using SETX, USP11, SPRTN and actin antibodies (left panel). SETX band intensities were normalized to actin and presented as fold increase as compared to C318S cells (right panel). Data are the average of five biological repeats and presented as mean ± SD. p-Values calculated using two-tailed Student's t-test. **h** Total RNA was extracted from USP11-knockout clones at passage 2–7 (sgRNA Cl-1 and 2) and control cells (Cont), reverse transcribed to cDNA, and USP11 and SETX transcripts were quantified by qPCR. Data are the mean of three biological repeats ± SD, normalized to actin transcript levels and presented as % reduction compared to controls. ns; p > 0.05, two-tailed Student's t-test. **i** Parental HEK-293 (Con) and USP11 sgRNA Cl-2 cells were incubated with cycloheximide (CHX). The expression level of SETX, USP11, TDP1 and Tubulin was analysed by immunoblotting. The images are representative of three biological repeats. **j** The band intensity of SETX and TDP1 following incubation with CHX was quantified from three biological repeats, normalized to tubulin and then presented as fold reduction of the mean ± SD compared to untreated samples; p > 0.05, two-tailed Student's t-test.

reduced SETX protein levels (Fig. 5g). We next examined whether endogenous KEAP1 interacts with SETX. Under unperturbed conditions, we observed a positive PLA signal, which was significantly increased following R-loop induction by CPT (Fig. 5h). The interaction between KEAP1 and SETX was transcription-dependent as the PLA signal was reduced by pre-incubation with the transcription inhibitor, AMN (Fig. 5h). Together, we conclude that SETX might be a substrate for KEAP1 E3 ubiquitin ligase activity.

**KEAP1 opposes USP11 to control SETX protein levels**. We next examined whether KEAP1 regulates SETX proteostasis by controlling its ubiquitination. Overexpression of KEAP1 led to an increased ubiquitination GFP-SETX$^{1–667}$, full-length FLAG-SETX and endogenous full-length SETX (Fig. 6a and Supplementary Fig. 5d, e). This was specific to KEAP1 as overexpression of the catalytic subunit of another E3 ubiquitin ligase, RNF168$^{1–110}$, did not increase SETX ubiquitination (Supplementary Fig. 5e). To test whether USP11 and KEAP1 act in an opposing manner to regulate SETX ubiquitination, we depleted USP11 and KEAP1 separately and together (Supplementary Fig. 5f), and examined SETX ubiquitination using denaturing pulldowns. Depletion of KEAP1 was sufficient to reduce the increased levels of SETX ubiquitination in USP11-deficient cells (Fig. 6b), suggesting that KEAP1 and USP11 antagonistically regulate SETX ubiquitination. Consistently, depletion of KEAP1 reduced the increased levels of R-loops in USP11-deficient cells (Fig. 6c). Furthermore, overexpression of KEAP1 increased the levels of K48-ubiquitinated SETX, as measured by denaturing pulldowns (Fig. 6d). These data are consistent with the reduced SETX protein levels following KEAP1 overexpression (Fig. 5g) and the increased SETX K48-ubiquitination following USP11 depletion (Fig. 4h). Together, we conclude that KEAP1 opposes USP11 to regulate SETX ubiquitination and proteostasis.

**Loss of USP11 reduces SETX binding to KEAP1 promoter and increases R-loops**. We next wondered why loss of USP11 reduces KEAP1 transcript and protein levels. First, we examined USP11

binding to established R-loop loci using chromatin immunoprecipitation (ChIP). USP11 was bound to R-loop-positive loci such as *28S*, *R7*, *ING3*, *Actin*, *EGR1* and R-loop-negative loci such as *MYADM-neg*, *SNRPN-neg* (Supplementary Fig. 6a), which is consistent with the broad roles of USP11 on chromatin$^{43,51}$. However, USP11 occupancy at R-loop-positive loci, but not R-loop-negative loci, was reduced following SETX depletion (Fig. 7a). This observation suggests that SETX promotes enrichment of USP11 at R-loops. Of note is the observed USP11 binding to *KEAP1* promoter, which is also reduced upon SETX depletion (Fig. 7a). To test whether SETX increases USP11 binding to chromatin globally, we performed nuclear fractionation assays. Depletion of SETX did not reduce USP11 levels in the insoluble chromatin fraction (Supplementary Fig. 6b), suggesting that SETX regulates USP11 chromatin enrichment at specific genomic sites and not globally. Next, we examined whether SETX binds to the same loci. As expected, SETX was enriched at R-loop-positive, but not -negative, loci (Supplementary Fig. 6c). Notably, we also observed a significant enrichment of SETX at the *KEAP1* promoter. Next, we examined whether enrichment of SETX on chromatin is USP11-dependent. Loss of USP11 reduced SETX binding to promoters (*KEAP1*, *EGR1*) and gene bodies (*ING3*) but not 3′ gene ends (*28S*, *R7*, *Actin*) (Fig. 7b and Supplementary Fig. 6d). The reduced SETX binding at *KEAP1* promoter in USP11-deficient cells suggest that perturbations in R-loops in this region may explain the reduced *KEAP1* transcripts upon USP11 loss. Consistently, loss of USP11 increased R-loop levels at the *KEAP1* promoter (Fig. 7c). Together, we conclude that loss of USP11 not only increases SETX K48-ubiquitination and protein turnover but also reduces the binding of the remaining pool of SETX to R-loop loci including the *KEAP1* promoter, leading to R-loop accumulation.

**R-loops formed following USP11 deficiency are converted to double-strand breaks**. We next examined whether the increased R-loops upon USP11 loss are converted to double-strand breaks. To test this, we performed γH2AX ChIP-qPCR using the same primers used for DRIP (Fig. 7d). Loss of USP11 increased γH2AX

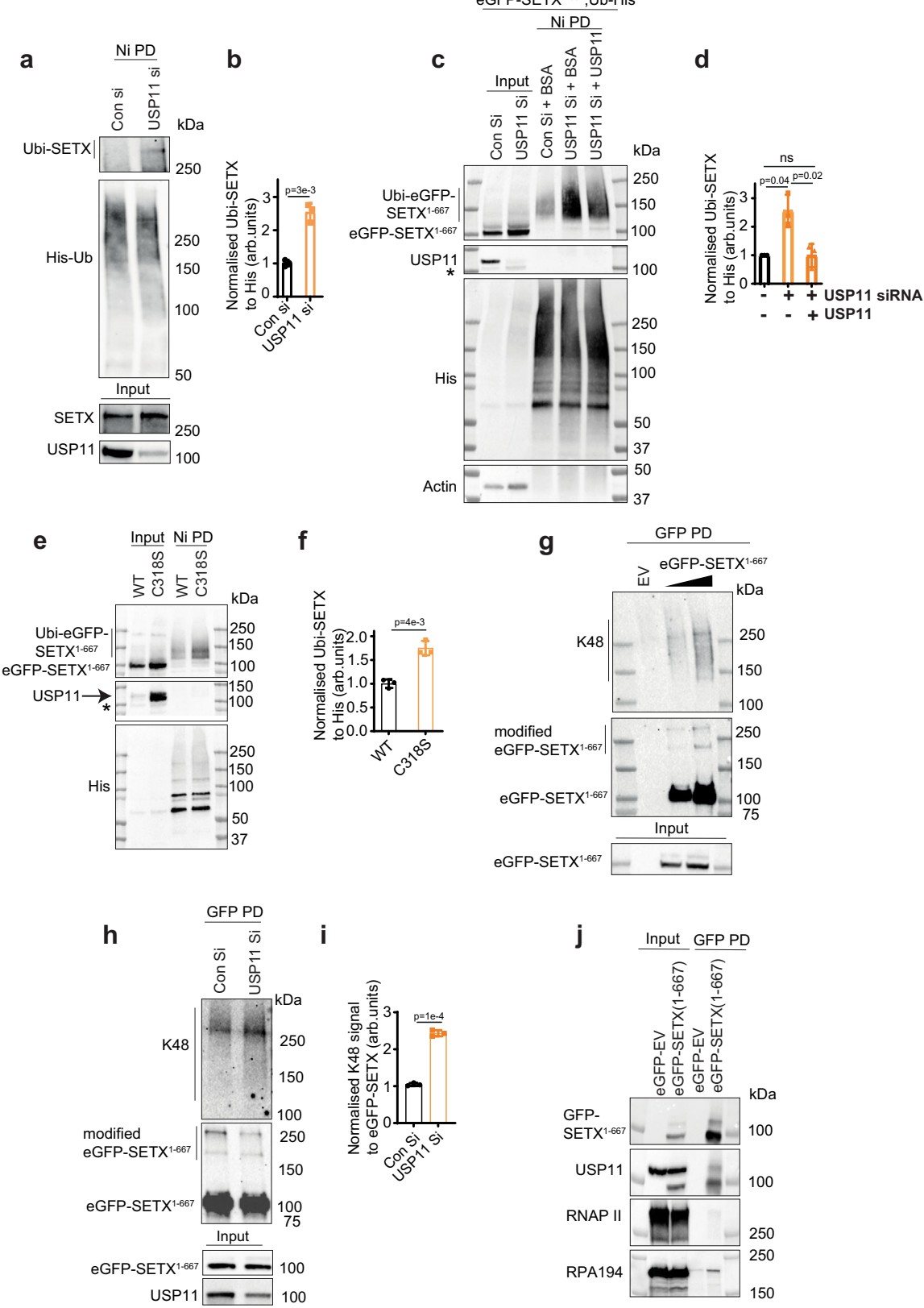

in both R-loop-positive and -negative loci, which is consistent with the role of USP11 in homologous recombination[39,41]. Importantly, however, the accumulation of γH2AX in USP11-depleted cells was reduced by RNase H1 only in R-loop-positive, but not R-loop-negative, loci (Fig. 7d). We therefore reasoned that the R-loops that accumulate in USP11-deficient cells might

be processed by endonucleases to double-strand breaks. XPF is a DNA repair structure-specific flap endonuclease that has been reported to cleave R-loops to double-strand breaks[52,53]. Depletion of USP11 increased XPF occupancy at R-loop-positive loci including the *KEAP1* promoter, in an RNase H1-dependent manner (Fig. 7e). Furthermore, USP11 depletion increased

**Fig. 4 USP11 regulates SETX ubiquitination. a, b** Control (Con si) and USP11-depleted HEK-293 cells (USP11 si) were transfected with Ub-His plasmid, treated with MG132 overnight (PI), lysed and subjected to nickel pull-down under denaturing conditions to purify ubiquitinated proteins. Samples were fractionated by SDS-PAGE and analysed by immunoblotting using anti-SETX, USP11 and His antibodies. **a** The band intensities of Ubi-SETX were normalized to His-Ub and presented as fold increase of SETX ubiquitination compared to controls (**b**). Data are the average ± SD from three biological repeats. *p*-Values calculated using two-tailed Student's *t*-test. **c, d** Purification of Ubi-eGFP-SETX$^{1-667}$ was conducted as described above followed by incubation of the nickel beads with BSA or recombinant USP11 in a deubiquitination buffer. Samples were fractionated by SDS-PAGE and analysed by immunoblotting using anti-GFP, USP11, His and actin antibodies. * denotes a nonspecific band (**c**). The band intensities of Ubi-eGFP-SETX$^{1-667}$ were normalized to His-Ub and presented as fold increase of SETX ubiquitination compared to controls (**d**). Data are the average ± SD from three biological repeats. ns; *p* > 0.05, two-tailed Student's *t*-test. **e, f** USP11 sgRNA Cl-2 cells expressing wild-type USP11 (WT) or the catalytically inactive USP11$^{C318S}$ mutant (C318S) were transfected with plasmids encoding eGFP-SETX$^{1-667}$ and Ub-His; and ubiquitinated eGFP-SETX$^{1-667}$ was purified using Nickel pull-down under denaturing conditions. * denotes a nonspecific band (**e**). The band intensities of Ubi-eGFP-SETX$^{1-667}$ were normalized to His-Ub and presented as fold increase of SETX ubiquitination compared to WT (**f**). Data are the average ± SD from three biological repeats. *p*-Values calculated using two-tailed Student's *t*-test. **g** HEK-293 cells expressing an empty vector-eGFP (EV) or eGFP-SETX$^{1-667}$ were lysed under denaturing conditions and subjected to GFP pull-down to purify eGFP-SETX$^{1-667}$. Samples were fractionated by SDS-PAGE and analysed by immunoblotting using anti-GFP and K48 antibodies. The images are representative of three biological repeats. **h, i** HEK-293 cells expressing eGFP-SETX$^{1-667}$ were transfected with control siRNA (Con Si) or USP11 siRNA (USP11 Si), lysed and subjected to GFP pull-down under denaturing conditions to purify eGFP-SETX$^{1-667}$. Samples were fractionated by SDS-PAGE and analysed by immunoblotting using anti-GFP, K48 and USP11 antibodies (**h**). The intensity of K48 signal was normalized to pulled down eGFP-SETX$^{1-667}$ signal and presented as fold increase of SETX K48-ubiquitination in USP11 Si cells compared to controls (**i**). Data are the average ± SD from three biological repeats. *p*-Values calculated using two-tailed Student's *t*-test. **j** HEK-293 cell lysates expressing an empty vector-eGFP (EV-GFP) or eGFP-SETX$^{1-667}$ were subjected to GFP pull-down and analysed by immunoblotting using antibodies against GFP, USP11, RNA polymerase II (RNAPII) and RNA polymerase I (RPA-194).

R-loops (Fig. 7f), XPF occupancy and γH2AX (Fig. 7g) at two established common fragile sites (*FRA16* and *FRA3B*)[54], which was again RNase H1-dependent. Taken together, we conclude that USP11 deficiency increases SETX ubiquitination, reduces SETX protein levels and binding to R-loops at specific genomic sites including *KEAP1* promoter and common fragile sites, which leads to formation of R-loop that are converted to double-strand breaks.

## Discussion

We describe a series of experiments defining USP11 as a DUB for SETX and thus an R-loop regulator. First, using multiple orthogonal methods, we show that loss of USP11 increases nuclear and nucleolar R-loops in human cells. We then demonstrate that USP11 and SETX are epistatic in terms of R-loop regulation and cell survival after treatment with R-loop inducing agents. We further show cellular physical proximity of endogenous USP11 and SETX, which is increased by R-loop induction in a transcription-dependent manner. USP11 depletion increased SETX K48-ubiquitination and SETX protein turnover. These effects were dependent on USP11 DUB activity as shown by experiments using a catalytically inactive mutant. The adaptation of USP11-knockout CRISPR cells and restoration of SETX protein levels in passage-dependent manner led to the identification of the role of KEAP1 in SETX regulation.

We provide evidence that KEAP1 interacts and ubiquitinates SETX with K48-ubiquitin chains, which targets SETX for degradation. We further show by DRIP-qPCR and ChIP-qPCR that loss of USP11 increases R-loop accumulation at *KEAP1* promoter, which triggers double-strand breaks, providing a plausible explanation for the culture adaptation and downregulation of *KEAP1* transcription in USP11-deficient cells. The down-regulation of *KEAP1* transcript levels was only observed in aged USP11-sgRNA clones (Fig. 5e, f), which is consistent with a model in which passage-dependent accumulation of R-loops caused by USP11 loss and subsequent SETX degradation triggers increased DNA damage (Fig. 2h) and accelerated ageing (Supplementary Fig. 5c), and that cells effectively adapt by down-regulation of *KEAP1*. This mode of regulation is not unprecedent. USP11 has been recently shown to positively regulate the stability of the master regulator of oxidative-stress response, NRF2[55], which is also negatively regulated by KEAP1[56]. Loss of USP11

increased ubiquitination of NRF2 and its subsequent degradation, which led to downregulation of transcription of NRF2-target genes[55]. Depletion of NRF2 was also shown to reduce transcription of *KEAP1*, resulting in low KEAP1 steady-state levels[57].

Our data support a model in which KEAP1 and USP11 antagonistically control ubiquitination of SETX to regulate its protein levels and consequently regulate R-loop homeostasis (Fig. 7h and Supplementary Fig. 6e). This is reminiscent to previous findings whereby USP11 and KEAP1 were also reported to antagonistically control the stability of PALB2[41] and NRF2[55,58]. The increased global binding of SETX to both USP11 and KEAP1 following R-loop induction by CPT is intriguing. We suggest the extent of this binding is adjusted to favour more binding to USP11, thus increasing SETX levels, or more binding to KEAP1, thus reducing SETX level, at different genomic loci. This layer of spatial regulation would ensure a fine balance between physiological R-loops that are required to promote transcription vs. pathological R-loops, which cause genomic instability. A genome-wide DRIP and ChIP-sequencing approach will address this idea. Nevertheless, pharmacological or pathological perturbations of this balance will have detrimental cellular consequences. For example, mitoxantrone is a small molecule inhibitor of USP11, which has been shown to be effective in cancer cells over-expressing USP11[59]. Our clonogenic cell survival data show that USP11-deficient cells are hypersensitive to R-loop-inducing agents such as CPT. Thus, a combination of CPT and mitoxantrone may offer a new targeted approach to treat certain cancers. Reduction of SETX levels via USP11 inhibition could also overcome cancer resistance caused by hypoxia[27]. Furthermore, as USP11 regulates SETX protein stability, mitoxantrone could be used to treat disorders where SETX is overexpressed or where its activity and levels are pathogenic. ALS4 is a neurodegenerative disorder caused by a gain-of-function SETX mutation[47]. Moderate, but prolonged, depletion of USP11 could lead to down-regulation of the mutated SETX, which in turn could alleviate or delay symptoms of ALS4. Furthermore, disorders characterized by an excessive R-loop accumulation with no reported link with USP11-SETX-KEAP1 axis, such as the most common form of familial ALS that is caused by *C9orf72* expansions[16,60,61], could potentially benefit from USP11 activators or KEAP1 inhibitors[62].

The majority of chromosomal breaks observed in USP11-deficient cells were caused by aberrant R-loop accumulation (Fig. 2h) and the

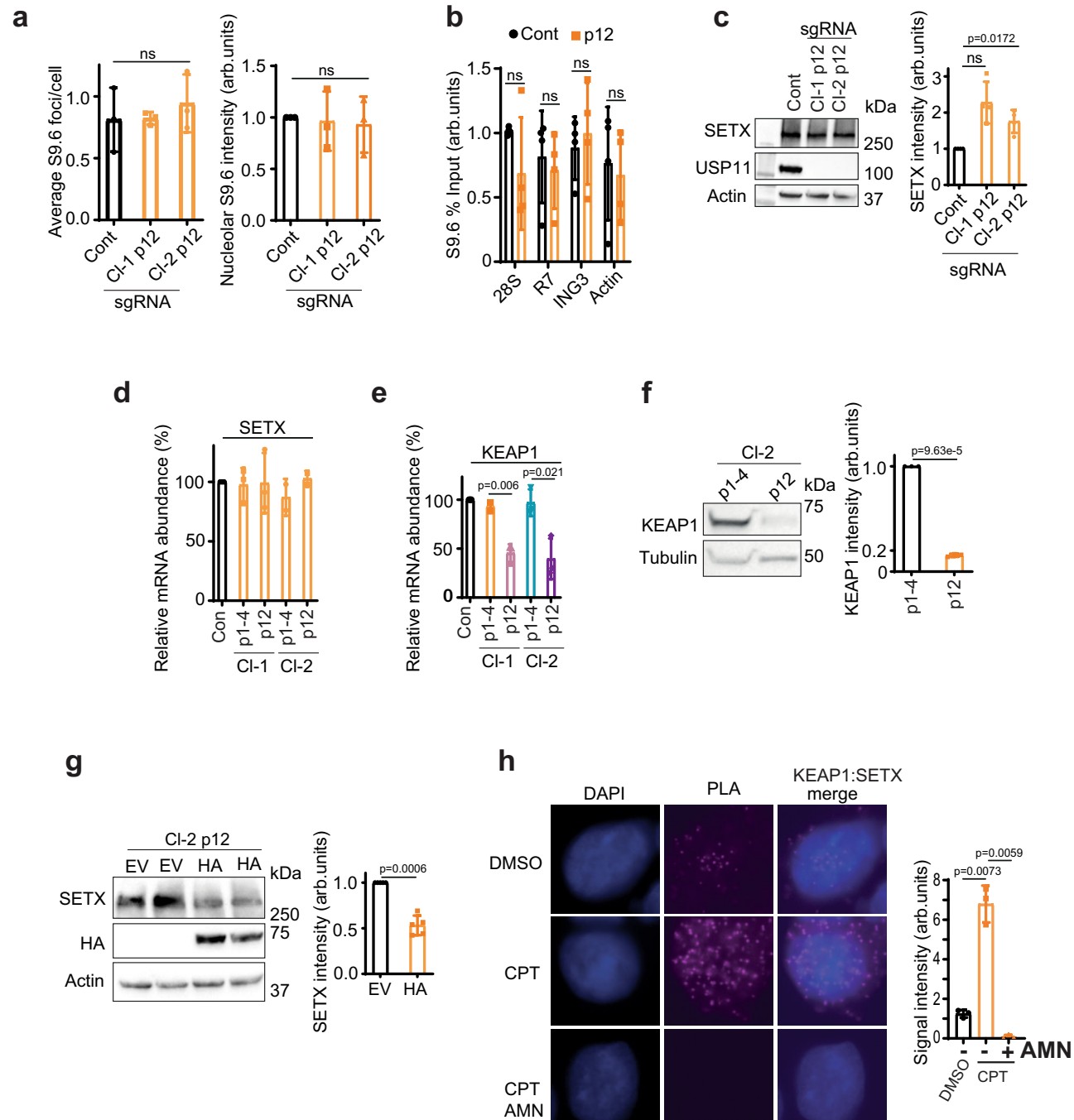

reported hypersensitivity of USP11-deficient cells to PARP inhibitors[39,41] was reduced by RNase H1 overexpression (Fig. 2i). This is consistent with recent reports implicating R-loops in the cellular response to PARP inhibitors[63] and with reports showing that the hypersensitivity of USP11-deficient cells to the PARP inhibitor, olaparib, could be overcome by depletion of KEAP1[41]. Therefore, perturbations in R-loop homeostasis are likely playing a key role in the evolving resistance to PARP inhibitors.

We further show that USP11 depletion reduces SETX enrichment at *KEAP1* promoter and increases R-loop accumulation, the double-strand break marker γH2AX and the endonuclease XPF, at the same genomic sites. These observations support a model in which loss of USP11 not only increases SETX K48-ubiquitination and reduces SETX protein stability but also reduces the binding of the remaining SETX pool to the *KEAP1* promoter. This leads to

R-loop accumulation, which are then processed by XPF to double-strand breaks, resulting in adaptation by transcriptional downregulation of *KEAP1*. This model is not likely to be restricted to *KEAP1*, as it was also true at two common fragile sites, which is consistent with published findings linking USP11 to genomic instability and carcinogenesis[64,65].

Loss of USP11 reduced SETX binding to promoters (*KEAP1*, *EGR1*) and gene bodies (*ING3*) but not transcription end sites (*28S*, *R7*, *Actin*) (Fig. 7b). These findings not only provide a plausible explanation for the culture adaptation by down-regulation of *KEAP1* transcription but are also exciting, because SETX was previously shown to be recruited by BRCA1 to transcription termination sites to resolve R-loops[66]; however, how SETX is recruited to promoter-proximal regions and gene bodies remain unknown. Our ChIP-qPCR data identify USP11 as a

**Fig. 5 Aged USP11-knockout cells restore SETX and R-loop levels via downregulation of KEAP1. a** Parental HEK-293 cells (Cont) and USP11-knockout clones 1 and 2 (Cl-1 and Cl-2) at passage 1–14 (p12) were examined for R-loop levels using S9.6/nucleolin immunostaining. The average number of S9.6 foci/cell (left panel) and total nucleolar fluorescence normalized to control cells (right panel) were calculated from three biological repeats, each containing at least 100 cells and presented as average ± SD. ns; $p > 0.05$, two-tailed Student's $t$-test. **b** Lysates from Control (Cont) and USP11-knockout clone-2 at passage 12–15 (p12) were subjected to DNA/RNA immunoprecipitation (DRIP) using S9.6 antibodies. Quantitative PCR was conducted using primers targeting nucleolar (*28S* and *R7*) and nuclear (*ING3* and *actin*) loci. The data represent the average ± SD from four biological repeats. Raw % input values are shown in Supplementary Fig. 5b. ns; $p > 0.05$, two-tailed Student's $t$-test. **c** Lysates from parental HEK-293 cells (Cont) and USP11-knockout clones 1 and 2 (Cl-1 and Cl-2) at passage 12–14 (p12) were fractionated by SDS-PAGE and analysed by immunoblotting (left panel). SETX band intensities were normalized to actin and presented as fold increase compared to control cells (right panel). Data are the mean of four biological repeats ± SD. ns; $p > 0.05$, two-tailed Student's $t$-test. **d, e** Total RNA was extracted from young (p1–4) and aged (p12–15; p12) USP11 sgRNA clones, reverse transcribed to cDNA and transcript levels of SETX and KEAP1 were quantified by qPCR. SETX and KEAP1 mRNA levels were first normalized to actin and then presented as % change compared to levels in control cells. Data are the mean ± SD from three biological repeats. $p$-Values calculated using two-tailed Student's $t$-test. **f** Lysates from young (p1–4) and aged (p12–15, p12) USP11 sgRNA clone-2 (Cl-2) were fractionated by SDS-PAGE and analysed by immunoblotting (left panel). KEAP1 band intensities were normalized to tubulin and presented as fold decrease compared to p1–4 (right panel). Data are the mean of three biological repeats ± SD. $p$-Values calculated using two-tailed Student's $t$-test. **g** Lysates from USP11-knockout clone-2 (Cl-2) at passage 12 were transfected with empty vector (EV) or HA-KEAP1 (HA) plasmids and fractionated by SDS-PAGE for immunoblotting analysis. Two biological repeats are shown (left panel). SETX band intensities were normalized to actin and presented as fold decrease compared to EV (right panel). Data are the mean of five biological repeats ± SD. $p$-Values calculated using two-tailed Student's $t$-test. **h** MRC-5 cells treated with mock (DMSO) or α-amanitin (AMN) overnight followed by 10 min incubation with 25 μM CPT were subjected to proximity ligation assay using antibodies against endogenous KEAP1 and SETX. Representative images are shown, scale bar is equal to 10 μm (left panel). The signal intensity was measured using ImageJ (right panel). Data are the mean ± SD from three biological repeats. $p$-Values calculated using two-tailed Student's $t$-test.

putative candidate fulfilling this function by recruiting SETX to non-transcription end sites. Whether this model is true and to what extent it is impacted across the lifespan and by ageing remain to be determined[67]. We also note that human SETX has not been shown to be involved in RNAPI termination, unlike its *S. cerevisiae* orthologue Sen1p[50]. Here, we demonstrate that ectopically expressed SETX[1–667] immunoprecipitates with RNAPI large subunit, RPA-194, suggesting a possible role for SETX in human RNAPI termination. Further, we show that both USP11 and SETX steady-state levels are increased as cells enter the S phase, which is consistent with the yeast data whereby Sen1p was upregulated in S phase and degraded in G1 by the Ubiquitin-Proteasome System[49], and with human data showing enrichment of R-loops in S phase compared to G1[68].

Finally, this report provides evidence and mechanistic explanation for how cellular phenotypes obtained by genome editing are subject to transcriptional adaptation. It opens new avenues of research that involves comparative investigation of young vs. aged gene-edited cells and organisms to uncover new genetic interactions.

In summary, we employed a genetic screen and took advantage of a cellular adaptation model to define a ubiquitination-dependent mechanism for R-loop regulation by SETX, which is controlled by the opposing activities of USP11 and KEAP1 with broad applications for cancer and neurological disease.

## Methods

**Cell lines and cell culture.** MRC-5, HEK-293 and HEK-293 CRISPR cell lines were grown as monolayers in minimum essential media (MEM) supplemented with 10% fetal calf serum (FCS) and 1% L-Glutamine in 5% $CO_2$ incubators at 37 °C. U2-OS cell line was grown in McCoy's 5A media (modified) supplemented with 10% FCS in 5% $CO_2$ incubators at 37 °C. All cells were routinely tested negative for mycoplasma.

**Antibodies, siRNAs, primers and plasmids.** All information regarding antibodies, siRNAs, primers and plasmids is listed in Supplementary Tables 1–4. siRNA transfections were carried out at a final 50 nM concentration unless otherwise stated.

**Generation of USP11 KO clones.** Small guide RNA against exon 1 of USP11 was designed—forward 5′-CAC CGA GAA CGG ACG GCG ATG GCG A-3′, reverse 5′-AAA CTC GCC ATC GCC GTC CGT TCT C-3′—and cloned into pSpCas9n(BB)-2A-GFP. HEK-293 cells were transfected with pSpCas9n(BB)-2A-USP11 sgRNA-GFP. Serial dilution was used to obtain homogenous colonies,

which were screened by immunoblotting with anti-USP11 antibody. Successful USP11-knockout (KO) clones were confirmed by DNA sequencing. USP11 sgRNA clones 1 and 2 were transfected with either wild-type FLAG-HA-USP11 or catalytically inactive FLAG-HA-USP11[C318S] plasmids and subsequently selected with puromycin to form stable populations expressing exogenous USP11.

**CHX chase experiment.** $2.0 \times 10^5$ USP11-sgRNA clone-2 or HEK-293 cells were seeded in six-well plates, (if required) transfected with USP11 siRNA or control siRNA to a final concentration of 25 nM, left for 48 h and treated with 50 μg/ml CHX and 5 μM MG132 for 24 h. Next, cells were lysed in 50 mM Tris-HCl pH 8, 40 mM NaCl, 2 mM $MgCl_2$, Triton (0.5% v/v), 1× Protease Inhibitor, 20 mM NEM, 250 units of Basemuncher (Expedeon) and analysed by immunoblotting. SETX band intensity was first normalized to actin and then presented as fold reduction compared to untreated samples.

**Reverse-transcription qPCR.** RNeasy® Mini Kit (Qiagen, Valencia, CA) was used to purify total RNA from HEK-293 cells. Total RNA was quantified (NanoDrop, ThermoFisher) and 5 μg were used immediately to synthesize cDNA (High Capacity RNA-to-cDNA Kit, catalogue number: 4387406, ThermoFisher). Twenty microlitres of the total reaction volume contained 1× Reverse Transcription Buffer, 4 mM dNTP Mix, 1× Reverse Transcription Random Primers, 1× MultiScribe™ Reverse Transcriptase and 5 μg of total RNA. The thermocycling conditions were set to 25 °C for 10 min, 37 °C for 120 min and 85 °C for 5 min. cDNA samples were diluted 1 : 10 in ddH₂O. Five microlitres of diluted cDNA was mixed with 1× SensiMix™ SYBR® & Fluorescein reaction mix (containing 3 mM $MgCl_2$) (Bioline, UK) and 0.7 nM primer pair in 20 μl total reaction volume. At this stage, all experimental and reference samples were prepared manually in duplicates. Thermocycling conditions were set to 95 °C for 10 min, which was followed by 45 cycles of 95 °C for 15 s, 55 °C for 15 s and 72 °C for 30 s. Rotor-Gene 6000 series Software 1.7 (Corbett Life Science) was used to determine the quantification cycle ($C_q$) value (which is a number at which the fluorescence signal is significantly above the background noise). Next, $C_q$ values were converted to copy numbers using standard curve method (four standard dilutions in duplicates). Gene expression was normalized to β-actin mRNA levels. Values in the control cells were treated as 100%.

**DNA/RNA immunoprecipitation.** DRIP for all the figures, except Supplementary Fig. 2c and Figs. 6c and 7c, was essentially performed as ChIP with minor modifications. For Supplementary Fig. 2c and Figs. 6c and 7c, a more recently published DRIP-qPCR protocol was followed[38]. Briefly, HEK-293 cells were lysed in standard ChIP lysis buffer for 10 min: 50 mM HEPES-KOH pH 7.5, 140 mM NaCl, 1 mM EDTA pH 8, 1% Triton X-100, 0.1% Sodium Deoxycholate and 0.1% SDS. Next, digests were sonicated for 7 min (30 s on, 30 s off) using Bioruptor Pico (Diagenode, Liege, Belgium). The sonicated samples were incubated with 28 μg of S9.6 antibodies for an hour at 4 °C. Meanwhile, Protein A Dynabeads (Thermo Fisher Scientific, Waltham, MA) were washed three times in standard RIPA buffer and added to the sonicated digests for overnight precipitation at 4 °C. The next day, samples were or were not on-bead digested with ec-RH (20 units, NEB, Ipswich, MA) for 2 h at 37 °C. Then, the beads were washed once with low-salt wash buffer (0.1% SDS, 1% Triton X-100, 2 mM EDTA, 20 mM Tris-HCl pH 8.0, 150 mM

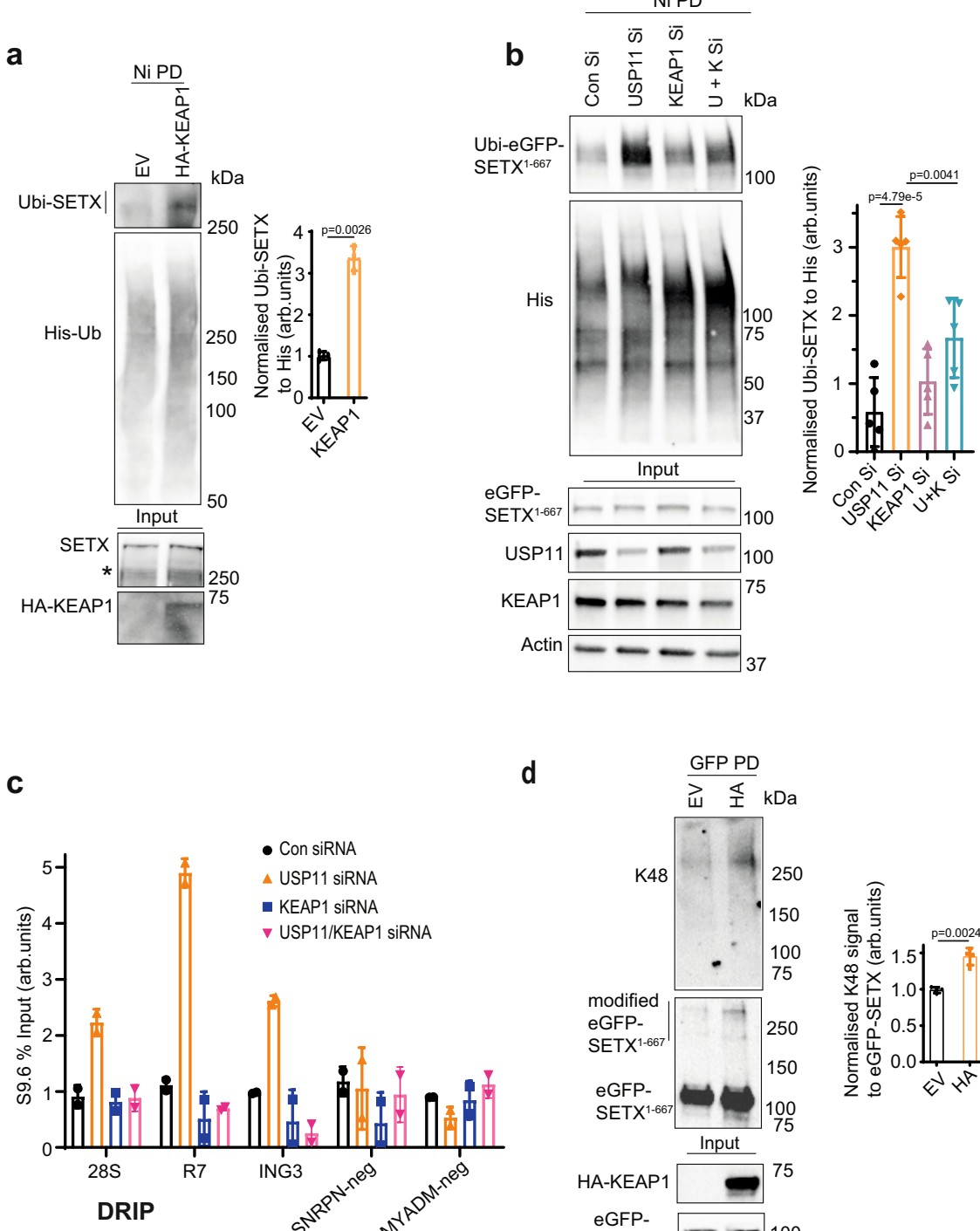

NaCl), once with high-salt wash buffer (0.1% SDS, 1% Triton X-100, 2 mM EDTA, 20 mM Tris-HCl pH 8.0, 300 mM NaCl) and once with Lithium Chloride wash buffer (0.25 M LiCl, 1% NP-40, 1% Sodium Deoxycholate, 1 mM EDTA, 10 mM Tris-HCl pH 8.0). Next, R-loops were eluted by adding 150 μL of elution buffer (1% SDS, 100 mM NaHCO₃) to the protein A Dynabeads and vortexed slowly for 15 min at 30 °C. Next, 150 mM NaCl, 150 μg/mL RNase A and 300 μg/mL proteinase K were added and incubated for 1 h at 37 °C. Next, R-loops were purified using standard phenol–chloroform protocol followed by ethanol precipitation. R-loops were resuspended in 50 μl RNase/DNase-free water. Two microlitres of the suspension were used for a qPCR reaction as described above for RT-qPCR. R-loop accumulation in each gene loci was determined as % input.

**Chromatin immunoprecipitation**. HEK-293 or USP11 sgRNA-2 HEK-293 cells were crosslinked for 10 min with 1% formaldehyde, quenched with 125 mM glycine

and lysed in ChIP lysis buffer (1% SDS, 5 mM EDTA, 50 mM Tris-HCl pH 8.1) for 10 min. Next, digests were sonicated for 7 min (7 cycles, 30 s on, 30 s off) using Bioruptor Pico (Diagenode, Liege, Belgium). One hundred micrograms of soni-cated samples were diluted 10× in ChIP dilution buffer (1% Triton X-100, 2 mM EDTA, 150 mM NaCl, 20 mM Tris-HCl pH 8.1) and incubated with an appropriate antibody overnight. Next day, 25 μl of Protein A or G Dynabeads (Thermo Fisher Scientific, Waltham, MA) were washed three times in the ChIP dilution buffer and added to the sonicated digests for 3 h immunoprecipitation at 4 °C. Next, the beads were washed once with low-salt wash buffer (0.1% SDS, 1% Triton X-100, 2 mM EDTA, 20 mM Tris-HCl pH 8.0, 150 mM NaCl), once with high-salt wash buffer (0.1% SDS, 1% Triton X-100, 2 mM EDTA, 20 mM Tris-HCl pH 8.0, 300 mM NaCl), once with Lithium Chloride wash buffer (0.25 M LiCl, 1% NP-40, 1% Sodium Deoxycholate, 1 mM EDTA, 10 mM Tris-HCl pH 8.0) and once with 1× TE buffer. Next, chromatin was eluted by adding 150 μL of elution buffer (1% SDS, 100 mM NaHCO₃) to the beads and vortexed slowly for 30 min at 55 °C. Next,

**Fig. 6 KEAP1 opposes USP11 to regulate SETX proteostasis. a** Empty vector (EV) and HA-KEAP1 (HA) overexpressing HEK-293 cells were transfected with Ub-His plasmid, lysed and subjected to nickel pulldown under denaturing conditions to purify ubiquitinated proteins. Samples were fractionated by SDS-PAGE and analysed by immunoblotting using SETX, HA and His antibodies. * Denotes a nonspecific band (left panel). The band intensities of Ubi-SETX were normalized to the bait His-Ub and presented as fold increase of SETX ubiquitination in HA-KEAP1 overexpressing cells compared to controls (right panel). Data are the average ± SD from three biological repeats. p-Values calculated using two-tailed Student's t-test. **b** Control (Con Si), USP11-, KEAP1- and double USP11/KEAP1 (U + K Si)-depleted HEK-293 cells were transfected with plasmids encoding eGFP-SETX$^{1-667}$ and Ub-His. Cell lysates were subjected to nickel pulldown under denaturing conditions to purify ubiquitinated proteins. Samples were fractionated by SDS-PAGE and analysed by immunoblotting using anti-GFP, USP11, His, KEAP1 and Actin antibodies (left panel). The band intensities of Ubi-eGFP-SETX$^{1-667}$ were normalized to His-Ub and presented as fold increase of SETX ubiquitination as compared to controls. Data are the average ± SD from four biological repeats (right panel). p-Values calculated using two-tailed Student's t-test. **c** Lysates from Control (Con siRNA), USP11-, KEAP1- and USP11/KEAP1-depleted cells (USP11/KEAP1 siRNA) were subjected to DNA/RNA immunoprecipitation (DRIP) protocol using S9.6 antibodies. Quantitative PCR was conducted using primers targeting nucleolar (28S and R7) and nuclear (ING3, SNRPN-neg, MYADM-neg) loci. SNRPN-neg and MYADM-neg loci are negative controls. Data represent the average ± range from two biological repeats. **d** HEK-293 cells expressing eGFP-SETX$^{1-667}$ were transfected with empty vector-HA (EV) or HA-KEAP1 (HA). Cell lysates were subjected to GFP pulldown under denaturing conditions to purify eGFP-SETX$^{1-667}$. Samples were fractionated by SDS-PAGE and analysed by immunoblotting using anti-GFP, K48 and HA antibodies (left panel). The intensity of K48 signal was normalized to pulled down eGFP-SETX$^{1-667}$ signal and presented as fold increase compared to EV. Data are the average ± SD from four biological repeats (right panel). p-Values calculated using two-tailed Student's t-test.

NaCl was added to 150 mM final concentration and samples were incubated for 24 h at 65 °C. Next, 150 µg/mL RNase A was added for 30 min at 37 °C and 300 µg/mL proteinase K was added for 2 h at 65 °C. Then, DNA was purified using standard phenol–chloroform protocol followed by ethanol precipitation. DNA was resuspended in 50 µl qPCR-quality water. Two microlitres of the suspension were used for a qPCR reaction as described above for RT-qPCR. Protein accumulation in each gene loci was determined as % input.

**DNA/RNA immunoprecipitation quantitative PCR.** DRIP-qPCR was conducted following a recently published protocol[38]. siRNA-transfected HEK-293 cells were grown in 15 cm dishes. They were lysed o/n in 37 °C in TE buffer containing 0.625% SDS and 62.5 µg/ml proteinase K. DNA was extracted following standard phenol–chloroform and ethanol precipitation methods. Post precipitation, DNA was spooled using a cut tip and subsequently digested overnight using a restriction enzyme cocktail (30 units of each: BsrGI, EcoRI, HindIII, SspI and XbaI). Next, 8 µg of digested DNA was incubated with 15 units of RNase-H enzyme for 4–6 h. Subsequently, 8 µg of digested DNA and 8 µg of RNase-H-treated DNA was incubated o/n with 20 µl of S9.6 antibody (ENH001, Kerafast) in DRIP binding buffer (10 mM sodium phosphate pH 7.0, 140 mM NaCl, 0.05% (v/v) Triton X-100, TE). Next, 100 µl of protein G beads were washed three times in DRIP binding buffer and added to the tubes from the previous step, and incubated for 2 h in 4 °C. Subsequently, the beads were washed three times in DRIP binding buffer and then R-loops were eluted in the elution buffer (50 mM Tris pH 8.0, 10 mM EDTA pH 8.0, 0.5% (v/v), 0.456 mg/ml proteinase K). The elution was carried out in a rotor at 55 °C for 45 min. Next, the eluate was subjected to standard phenol–chloroform extraction and ethanol precipitation. Obtained DNA was quantified by qPCR as described above. R-loop accumulation in each gene loci was determined as % input.

**Fractionation of nuclear fractions.** HEK-293 cells were lysed on ice in hypotonic buffer (20 mM HEPES pH 8.0, 10 mM KCl, 1 mM MgCl$_2$, 20% glycerol, 0.1% Triton X-100) for 10 min, to remove cytoplasmic proteins; next, cells were lysed in hypertonic buffer (20 mM HEPES pH 8.0, 1 mM EDTA, 400 mM NaCl, 20% glycerol, 0.1% Triton X-100) for 20 min, to isolate soluble nuclear fraction, and finally lysed in insoluble buffer (20 mM TRIS pH 8.0, 150 mM NaCl, 1% SDS, 1% NP-40, 10 mM iodoacetamide) for 50 min to collect insoluble nuclear fraction.

**S9.6 immunofluorescence.** Next, 8 × 10$^4$ MRC-5 cells were plated on round 13 mm glass coverslips followed or not by siRNA transfection. Following 48 h, cells were treated with 50 µM AMN for 15 h, 50 µM MG132 for 2 h or 25 µM CPT for 10 min. Dimethyl sulfoxide (DMSO) served as a negative control. Next, cells were washed in ice-cold phosphate-buffered saline (PBS), fixed with methanol : acetone (1 : 1) solution for 10 min at −20 °C. This was followed by three rounds of PBS washes and an incubation with 3% filtered bovine serum albumin (BSA) for 30 min at room temperature. Next, primary antibodies were added for an hour at room temperature, which was followed by three rounds of PBS washes and secondary antibodies + DAPI (4′,6-diamidino-2-phenylindole) incubation for an hour. Next, three PBS washes were carried out and the coverslips fixed on 26 × 76 mm microscope slides using VectaShield mounting medium H-1000 (Vector Laboratories, Burlingame, CA). Images were taken under Leica FW4000 Fluorescent Microscope (Leica Biosystems, Wetzlar, Germany) or Nikon confocal microscope system A1 (Nikon Instruments, Tokyo, Japan). All the images were processed and analysed in ImageJ in the same manner. Nucleolar and nuclear masks were generated for anti-nucleolin images using ImageJ. Nucleolar and nuclear intensities were corrected for the background noise.

**Proximity ligation assay.** Duolink™ PLA (Merck, Darmstadt, Germany) kit was used following manufacturer's instructions. For negative controls, the protocol was followed with an omission of one or both primary antibodies. In brief, 6 × 10$^4$ MRC-5 cells were plated on round 13 mm glass coverslips on day 1. On day 2, cells were treated with 50 µM AMN overnight as indicated. On day 3, cells were treated with 25 µM CPT for 10 min. DMSO was used as a negative control. Subsequently, cells were washed in ice-cold PBS, fixed with methanol for 5 min at −20 °C, washed again and incubated in PLA blocking buffer for 60 min at 37 °C. If cells were transfected with enhanced GFP (eGFP) plasmids, 20 min 4% paraformaldehyde at room temperature incubation followed by 3 min 0.2% Triton X-100 on ice incubation were used instead of methanol, in order to preserve eGFP signal. After blocking, cells were incubated for 1 h at room temperature with appropriate primary antibodies, followed by incubation with PLUS and MINUS PLA probes for 60 min at 37 °C, ligation mix for 30 min at 37 °C and amplification mix for 100 min at 37 °C. Finally, washed coverslips were fixed on 26 × 76 mm microscope slides using VectaShield mounting medium H-1000 (Vector Laboratories, Burlingame, CA). Images were taken under Leica FW4000 Fluorescent Microscope (Leica Biosystems, Wetzlar, Germany) and processed and analysed in ImageJ in the same manner.

**Clonogenic cell survival assay.** Four thousand siRNA-transfected MRC-5 cells were seeded in a 10 cm dish. Olaparib was added for 24 h, CPT for 1.5 h and formaldehyde for 3 h. Once the incubation time was over, the media was changed and cells were left for 7 days to grow. Media was aspirated, plates were air-dried and cells were fixed with 80% ethanol for 15 min. Then, ethanol was aspirated, plates were air-dried and cells were stained with 1% Methylene Blue for 1 h. Next, the plates were washed, air-dried and the colonies were counted. The number of colonies on untreated plates were treated as 100%. Each biological repeat was an average of three technical repeats.

**5-Ethynyl uridine staining.** Protocol for Click-iT® RNA Imaging Kit (Invitrogen, Carlsbad, CA) was used. Briefly, 8 × 10$^4$ cells were seeded on a 13 mm-round glass coverslip. Next, following cell starvation and AMN treatment, cells were fed FCS-supplemented media with 5-ethynyl uridine. This was followed by cell fixation, permeabilization, staining of the 5-ethynyl uridine and mounting. Images were taken under Leica FW4000 Fluorescent microscope (Leica Biosystems).

**GFP pulldown.** HEK-293 cells grown in 15 cm dishes were lysed on ice for 30 min in 200 µl of 20 mM HEPES (pH 7.4), 40 mM (NaCl), 2 mM MgCl$_2$, Triton (1% v/v), 1× Protease Inhibitor, 40 mM NEM and 250 units of Basemucher (Expedeon, San Diego, CA). Next, cells were spun down at 20,000 × g for 10 min at 4 °C. The supernatant was aspirated and diluted to 1 ml in 10 mM Tris-HCl pH 7.5, 150 mM NaCl and 0.5 mM EDTA. Twenty-five microlitres of GFP-Trap®_MA beads (Chromotek, Planegg-Martinsried, Germany) were washed three times in 10 mM Tris-HCl pH 7.5, 150 mM NaCl, 0.5 mM EDTA and added to the supernatants. The beads and supernatant were left to rotate for 2 h at 4 °C, which was followed by three washes in 10 mM Tris-HCl pH 7.5, 150 mM NaCl and 0.5 mM EDTA. Next, bound proteins were eluted by adding 1× SDS loading buffer and boiling at 95 °C for 10 min with periodic vortexing. Samples were run on a 4–15% precast gel (BioRad, Hercules, CA) and examined by immunoblotting.

**Pulldown under denaturing conditions.** HEK-293 cells grown onto 15 cm dishes were incubated with 25 µM MG132, for 2 h for USP11 depletion experiments or 10 µM MG132 overnight for KEAP1 overexpression, experiments prior to lysis, which was carried out in 500 µl of 50 mM Tris-HCl (pH 8), 40 mM NaCl, 2 mM MgCl$_2$, Triton (0.5% v/v), 1× Protease Inhibitor, 20 mM NEM and 250 units of

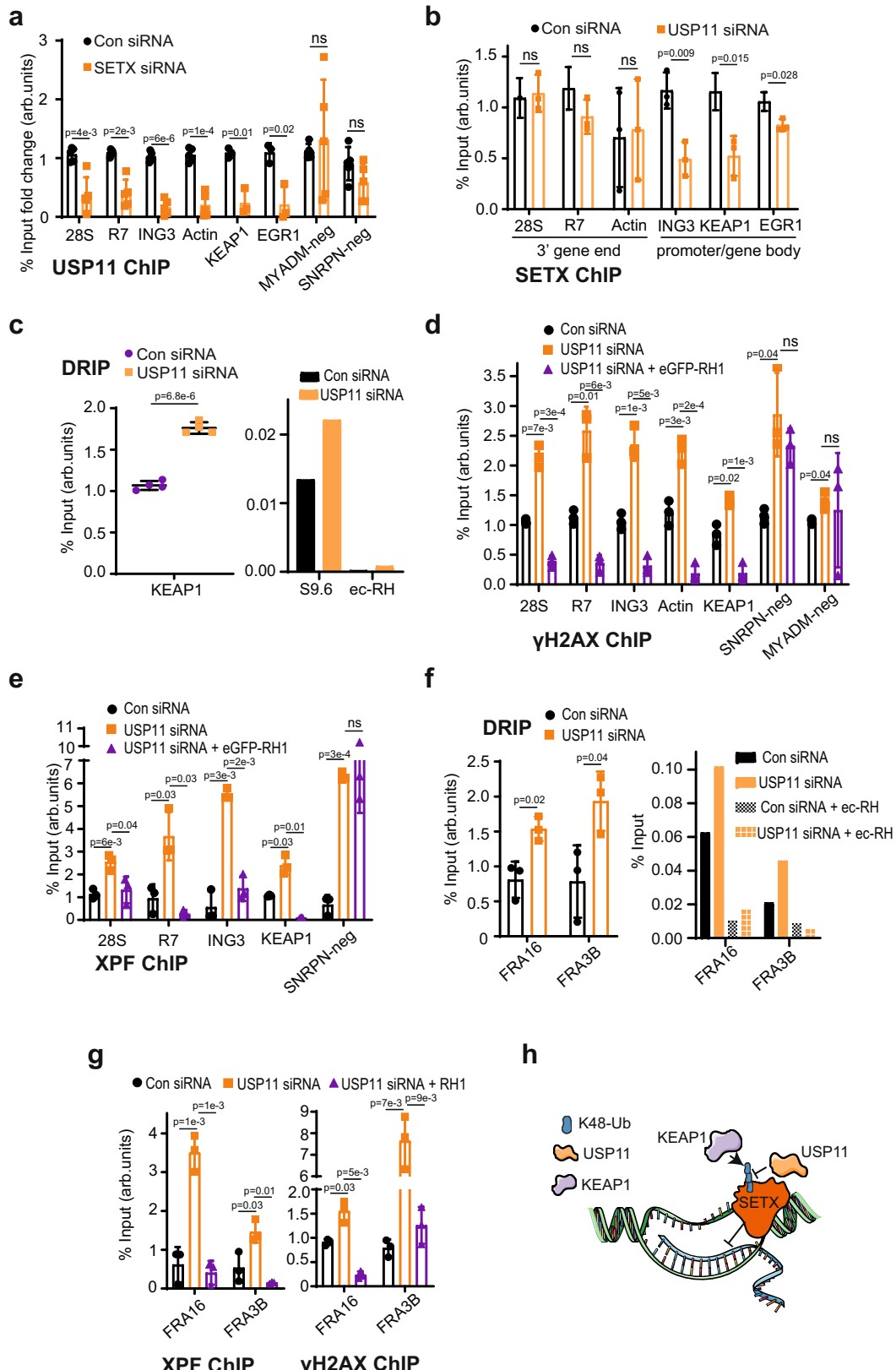

Basemucher (Expedeon). Post lysis, 1 volume of equilibration buffer (6 M guani-dine hydrochloride, 0.05% (v/v) Tween-20, 40 mM imidazole) was added to the lysates. Thirty microlitres of GFP-Trap®_MA beads (Chromotek, Planegg-Mar-tinsried, Germany) or His-Pur™ Ni-NTA magnetic beads (ThermoFisher) were washed twice in a wash buffer (8 M Urea, 0.05% (v/v) Tween-20, 50 mM imidazole,

PBS pH 8.0). Diluted cell lysates were combined with the washed beads and left to rotate for 1 h at room temperature. Next, the beads were washed three times in the wash buffer. Then, the beads were boiled in 1× SDS protein loading buffer for 10 min with periodic vortexing. Then, samples were loaded on a 4–15% precast gel (BioRad) and examined by immunoblotting.

**Fig. 7 R-loops formed upon USP11 loss are processed to double-strand breaks. a** Control siRNA (Con) or SETX siRNA-treated HEK-293 cells were subjected to a USP11 ChIP followed by qPCR using primers targeting nucleolar (*28S, R7*) and nuclear (*ING3, Actin, KEAP1, EGR1, SNRPN-neg, MYADM-neg*) loci and presented as % input fold change compared to Con. Data represent the average ± SD from five biological repeats. ns; $p > 0.05$, two-tailed Student's *t*-test. **b** Control siRNA (Con) or USP11 siRNA-treated HEK-293 cells were subjected to a SETX ChIP followed by qPCR using primers targeting nucleolar (*28S, R7*) and nuclear (*ING3, Actin, KEAP1, EGR1*) loci. Data represent the average ± SD from three biological repeats. ns; $p > 0.05$, two-tailed Student's *t*-test. **c** Lysates from Control (Con siRNA) and USP11-depleted cells (USP11 siRNA) were subjected to a DNA/RNA immunoprecipitation (DRIP) protocol using S9.6 antibodies. Quantitative PCR was conducted using primers targeting nuclear *KEAP1* locus. Pooled repeats (left panel) and raw % input values from a representative experiment are shown (right panel), and data represent the average ± SD from four biological repeats. *p*-Values calculated using two-tailed Student's *t*-test. **d** HEK-293 cells were transfected with Control siRNA, USP11 siRNA and eGFP-RNase H1 (eGFP-RH1), and subsequently subjected to a γH2AX ChIP followed by qPCR using primers targeting nucleolar (*28S, R7*) and nuclear (*ING3, Actin, KEAP1, SNRPN-neg, MYADM-neg*) loci. Data represent the average ± SD from three biological repeats. ns; $p > 0.05$, two-tailed Student's *t*-test. **e** HEK-293 cells were transfected with Control (Con) siRNA, USP11 siRNA and eGFP-RNase H1 (eGFP-RH1), and subsequently subjected to a XPF ChIP followed by qPCR using primers targeting nucleolar (*28S, R7*) and nuclear (*ING3, KEAP1, SNRPN-neg*) loci. Data represent the average ± SD from three biological repeats. ns; $p > 0.05$, two-tailed Student's *t*-test. **f** Lysates from Control (Con) siRNA and USP11-depleted cells (USP11 siRNA) were subjected to a DNA/RNA immunoprecipitation (DRIP) protocol using S9.6 antibodies. Quantitative PCR was conducted using primers targeting nuclear common fragile sites *FRA16* and *FRA3B*. In vitro, on-bead ec-RNase-H (ec-RH) treatment served as a signal validation control. Pooled repeats (left panel) and raw % input values from a representative experiment are shown (right panel), and data represent the average ± SD from three biological repeats. *p*-Values calculated using two-tailed Student's *t*-test. **g** HEK-293 cells were transfected with Control (Con) siRNA, USP11 siRNA and eGFP-RNase H1 (RH1), and subsequently subjected to a XPF ChIP (left panel) and γH2A.X ChIP (right panel) followed by qPCR using primers targeting nuclear common fragile sites *FRA16* and *FRA3B*. Data represent the average ± SD from three biological repeats. *p*-Values calculated using two-tailed Student's *t*-test. **h** A model depicting R-loop regulation by SETX-USP11-KEAP1 axis. SETX protein level is regulated via ubiquitination by KEAP1 and deubiquitination by USP11. We suggest that the extent of SETX binding to USP11 and KEAP1 is controlled to favour more binding to USP11, thus increasing SETX levels, or more binding to KEAP1, thus reducing SETX level, at distinct genomic loci. The spatial regulation and control of SETX levels would ensure a fine balance to favour physiological R-loops that are required to promote transcription and, at the same time, suppress pathological R-loops that cause genomic instability.

**In vitro deubiquitination of GFP-SETX**. Following nickel pull-down assay under denaturing conditions as described above, nickel beads were not boiled in 1× SDS buffer but split into two parts. One part was incubated with 2.88 µg/ml BSA, whereas the other with 2.88 µg/ml recombinant USP11 (Abcam, ab206019) for 4 h at 30 °C in the deubiquitination buffer (50 mM Tris-HCl pH 8.0, 50 mM NaCl, 1 mM EDTA, 1 mM dithiothreitol and 5% glycerol). Next, the beads were washed and boiled as described above.

**Double-thymidine block**. Fifty percent confluent HEK-293 cells were supplemented with 4 mM thymidine and were left to incubate for 18 h. Next, thymidine-containing media was aspirated and fresh media was added for 8 h, which was followed by another incubation with 4 mM thymidine for 18 h.

**DUB screen**. An ON-TARGET plus siRNA library for human DUBs (G-104705-05, GE Life Sciences) was employed in a 96-well format. The library contained 99 pools of 4 siRNAs against each of 99 human DUBs. The library was split between two 96-well plates for each biological repeat and 4 wells/plate containing control siRNAs were added. First, 50 µl of transfection mixture containing 0.3 µl DharmaFECT1 (Dharmacon, Lafayette, CO) were added to DUB/control siRNA pools in 96-well plates. Next, 4750 MRC-5 cells were added in 100 µl of MEM on top of the transfection mixtures. The plates were left for 48 h prior to a 10 min-long 25 µM CPT treatment, which was followed by S9.6 immunofluorescence assay. DMSO was used as a control to CPT treatment. Once DMSO/CPT-containing media was aspirated, cells were washed three time with ice-cold PBS, fixed with methanol:acetone (1:1) for 10 min at −20 °C and incubated with 3% filtered BSA for 30 min at room temperature. This was followed by S9.6 antibody incubation for an hour at room temperature. S9.6 was recycled, cells were washed three times with PBS and secondary antibody + DAPI solution was added for an hour. Then, cells were washed three times with PBS and stored in 100 µl of PBS in 4 °C. Images of cells were automatically taken using ImageXpress® Micro XLS Widefield High-Content Analysis System (Molecular Devices, San Jose, CA). All acquired images were automatically quantified by Custom Module of MetaXpress® software, which was designed to count an average number of S9.6 foci per nucleus. The software scored only large and bright S9.6 foci, which corresponded to nucleolar R-loops. Next, the S9.6 foci/cell counts were normalized against controls (control siRNA-treated cells). Detailed screen data for all 99 DUBs can be found in the Supplementary Data 1.

**Slot blotting**. Extracted genomic DNA (High Pure PCR Template Preparation Kit, Roche, Basel, Switzerland) was spotted in duplicates on a nitrocellulose membrane, which was subsequently air-dried, UV-crosslinked (0.12 J/m²) and immunoblotted using S9.6 and single-stranded DNA (ssDNA) antibodies. For ssDNA immunoblotting, one of the membranes was incubated for 10 min in denaturing (1.5 M NaCl, 0.5 M NaOH) and then 10 min in neutralizing solutions (0.5 M Tris-HCl pH 7.0, 1 M NaOH) prior to UV-crosslinking (0.12 J/m²).

**Statistical analysis**. Statistical analysis was conducted using Student's *t*-test for pair-wise comparisons. Two-tailed distribution and two-sample unequal variance settings were chosen. All data are presented as the average values of at least three biological repeats. Error bars represent SD unless otherwise stated. $p < 0.05$ was considered to be statistically significant.

**Reporting summary**. Further information on research design is available in the Nature Research Reporting Summary linked to this article.

## Data availability

The data that support this study are available from the corresponding author upon reasonable request. The raw DUB siRNA screen data generated in this study are provided in the Supplementary Information (Supplementary Data 1). Source data are provided with this paper.

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

## Acknowledgements
We thank Stephen Brown for help with the DUB screen, Stuart Wilson for the Flag-SETX construct and members of the El-Khamisy lab for useful discussions. This work was funded by a Wellcome Trust Investigator Award (103844) and a Lister Institute of Preventative Medicine Fellowship (137661). M.J. was additionally funded by a University of Sheffield PhD Studentship. Stock images used in Fig. 7h and Supplementary Fig. 6e were produced by Servier and obtained via bioicons.com under CC-BY 3.0 Unported (https://creativecommons.org/licenses/by/3.0/) license—the protein images were slightly modified. Stock image used in Fig. 1g came from bioicons.com under CC0 license.

## Author contributions
M.J. performed all experiments and data analysis. A.A. performed S9.6 IF in the U2-OS cells. M.J. and S.E.K. interpreted the data and wrote the manuscript. S.E.K supervised project with help from A.S.H.G. All authors edited the manuscript. S.E.K. conceived the study and managed the project.

## Competing interests
The authors declare no competing interests.
