## [Peer Review File · Nature Communications]

USP11 controls R-loops by regulating senataxin proteostasisReviewers' comments:

Reviewer #1 (Remarks to the Author):

Manuscript by Jurga et al describes a potentially interesting and new function of USP11 in limiting R-loops by regulating senataxin (SETX) protein stability in human cells. USP11 was uncovered by a DUB siRNA screen to inhibit R-loop signal at nucleoli. Subsequently, the authors show that USP11 depletion increases SETX ubiquitination and protein stability. Since SETX is a helicase that controls the level of R-loop formation, it makes sense that loss of USP11 leads to reduced SETX protein stability and more R-loop formation. Interestingly, prolonged loss of Usp11 by CRISPR/Cas9 deletion leads to an adaptation and restoration of SETX and R-loop levels by transcriptional downregulation of KEAP1, an E3 ubiquitin ligase. Finally, the authors show that overexpression of KEAP1 increases SETX ubiquitination, suggesting that KEAP1 is the ubiquitin ligase that targets SETX for degradation and this is being opposed by USP11. Although the model proposed by this study is compelling, the data falls way short in providing solid evidence of this link between USP11, KEAP1 and SETX, both biochemically and functionally.

Major points:

1) The direct physiological effect of USP11 on SETX ubiquitination is lacking. SETX expression is manipulated by ectopic overexpression so it is unclear whether USP11 directly acts on SETX. For example, when does USP11 associate with SETX (constitutive binding or only when RNA Pol I or Pol II is stalled, creating R-loops)? How does it associate or recognize SETX? What is USP11 doing to regulate SETX ubiquitination? If it isn't to deubiquitinate K48-linked SETX, then how is SETX degraded in a ubiquitin-dependent manner?

2) Similarly, the link between KEAP1 and SETX (and USP11) is unconvincing. The transcriptional downregulation of Keap1 upon continuous passaging of USP11 KO cells should lead to decrease in endogenous KEAP1 proteins (this is not shown) (Figure 6F). The reduction of SETX upon HA-KEAP1 overexpression is also unconvincing (Figure 6F). How is loss of USP11 causing Keap1 transcriptional inhibition? Is R-loop formation at Keap1 gene preventing its normal expression? How is Keap1 promoting SETX ubiquitination? Does Keap1 associate with SETX constitutively or only after SETX is required to be downregulated when R-loop levels go down? Does SETX need to be dynamically ubiquitinated and regulated by an E3 ligase and DUB for R-loop homeostasis? What happens when Keap1 is downregulated in the presence or absence of USP11?

3) The functional readout for R-loop formation is lacking. It is unclear what an increase in R-loop intensity in nucleoli or at nuclear DNA regions is telling us about transcription or genome integrity. The authors use nucleoli foci number and intensity as their functional readout but R-loops themselves are nothing special. There is a natural dynamics in R-loop formation and dissociation at promoter and 3-end regions of transcriptionally active genes. Does disruption of USP11 or SETX or KEAP1 reveal ssDNA or dsDNA breaks at these R-loop sites, and lead to genomic instability? Or does USP11 or KEAP1 play a role in transcriptional termination and gene function at transcriptionally active gene sites with higher intensity R-loops?

Reviewer #2 (Remarks to the Author):

Jurga et al propose a novel function for the de-ubiquitylating enzyme USP11 in regulating senataxin (SETX) ubiquitylation, protein stability and, as a consequence, R loop levels. After monitoring the specificity of the R loop signal detected with the S9.6 antibody, the authors performed an siRNA screen to identify de-ubiquitylating enzyme that could affect R loop levels. They identify and further characterize USP11 function. In particular, they show that USP11 interacts with and de-ubiquitylates SETX, thus controlling R loop levels. They corroborate these results in CRISPR-knockout (KO) cells. However, cells adapt after several passages by reducing the levels of KEAP1, a putative E3-ligase for SETX.

Overall, the amount of data generated is impressive. The first set of data (R loop phenotype) is quite convincing and it would be informative to further explore the well-established link between USP11 and the DNA damage. Is the DNA damage induction observed upon USP11 depletion (10.1074/jbc.M110.104745) mediated by R loop formation or by alternative mechanisms? The second part, in particular the impact of USP11 depletion on SETX turnover, is less convincing and needs additional controls. The ubiquitylation and interaction experiments have been mostly performed with the truncated SETX; what about the full-length? Due to the pleiotropic effect of the deubiquitylating enzymes, the final proof of the specificity of the observed phenotype would be rescue experiments in the USP11 KO with the WT or the USP11 defective for SETX binding. KEAP1 downregulation as an adaptation mechanism in the USP11 KO cells is intriguing, although very preliminary. Additionally, the authors show that USP11 is enriched at genes with elevated R loop levels (figure 7a, not described in the results section). This is an interesting finding and could deserve further exploration (i.e. Does USP11 accumulation depend on SETX?). Finally, it could be relevant to test whether AOA2- and ALS4-associated mutations of SETX affect the interaction with USP11.

Additional considerations:

- Fig. 1a-b. It would be good to show CPT 30min images as well. The authors state that CPT treatment does not change the morphology of the nucleoli, a quantification would help.
- Supp. Fig. 3: The authors use the cycloheximide chase data to support the idea that SETX turnover is faster in USP11-depleted cells, however the treatments also impact on USP11 protein levels. It may be worth repeating this experiment in the early KO cells. Are the upper bands in the SETX blot modified forms? Can the author comment on the proteasome inhibition? USP11 depletion is barely observed in this condition.
- Fig. 3: Is the full-length SETX ubiquitylated as well?
- Supp. Fig. 7a: showing in parallel wt control and non-complemented clones would be more convincing.
- Fig. 5b. As above, it would be good to have a wt control and the non-complemented clones for a direct comparison of SETX expression
- Fig. 5d. As above. Moreover, inputs are missing.
- Fig. 6g. It would help the reader mentioning in the text why the RNF1681-110 and not the full-length is used.

Reviewer #3 (Remarks to the Author):

It is well established that R-loops occur as part of normal transcription participating in pausing of transcription and termination events to regulate the overall process. However, why they play beneficial roles their persistence can lead to genome instability so it is necessary to have suitable mechanisms in place to prevent their accumulation. This is mediated by enzymes such as senataxin and RNase H which mediate their removal. The importance of such mechanisms is the occurrence of syndromes such as AOA2 and ALS4 associated with mutations in SETX the gene coding for senataxin. While the functioning of senataxin has received considerable attention over the last several years its regulation at the protein level has not been described. The present submission addresses this issue in that it describes how the deubiquitinase (Dub) USP11 regulates senataxin proteostasis to control the accumulation of R-loops. The work described in the submission is novel, very thorough and of interest to those involved in studying the DNA damage response but also to a wider audience working on transcription regulation and consequences of deficiency in cell function and disease such as cancer. Use of statistics clear and convincing.

Specific comments

1. This work provides new insight into how a specific Dub, USP11, for the helicase senataxin regulates the level of R-loops to minimize the risk of genome instability. It addresses this in a series of experiments by showing that deletion of USP11 results in ubiquitination and degradation of senataxin and as a consequence accumulation of R-loops.

2. The series of experiments designed to demonstrate this are well thought out and convincing.
3. In relation to Fig4a while the CRISPR –generated clones had no detectable USP11 the level of protein in the WT is also almost undetectable. This may be due to the high levels of overexpression of the catalytically inactive C318S mutant? It would be useful to have a blot with more convincing data for the WT. Also why is the mutant so highly expressed? How do you explain this by a positive-feed back loop?
4. In Fig 1 why the high background signal for S9.6 (R-loops) in the untreated control?
5. There is some confusion in the positions of 250 markers and setx. For example in Fig 2f. There are two setx bands one ubiquitinated and the other not . Where should the 250 marker be? Below the main band?
6. In Fig 6f we see two bands again. This time the 250 marker appears to coincide with the lower band. Is this setx or is it the upper band?
7. In further reference to Fig6f it is stated that this is “ Consistent with this model, overexpression of KEAP1 in aged USP11 knockout cells restored SETX protein to levels observed in young early passage cells (Fig. 6F).
Restoration of setx is not obvious when considering either the upper or lower bands?
8. Nucleolar enrichment of R-loops already described (referred to in Fig 1) should be referenced e.g Yeo et al 2014.
9. Can you comment further on cellular adaptation with respect in young versus old cell lines

Please find below our *point-by-point* response to reviewers' comments:

Reviewer 1

*Manuscript by Jurga et al describes a **potentially interesting** and new function of USP11 in limiting R-loops by regulating senataxin (SETX) protein stability in human cells. USP11 was uncovered by a DUB siRNA screen to inhibit R-loop signal at nucleoli. Subsequently, the authors show that USP11 depletion increases SETX ubiquitination and protein stability. Since SETX is a helicase that controls the level of R-loop formation, it makes sense that loss of USP11 leads to reduced SETX protein stability and more R-loop formation. Interestingly, prolonged loss of Usp11 by CRISPR/Cas9 deletion leads to an adaptation and restoration of SETX and R-loop levels by transcriptional downregulation of KEAP1, an E3 ubiquitin ligase. Finally, the authors show that overexpression of KEAP1 increases SETX ubiquitination, suggesting that KEAP1 is the ubiquitin ligase that targets SETX for degradation and this is being opposed by USP11. Although the **model proposed by this study is compelling**, the data falls short in providing solid evidence of this link between USP11, KEAP1 and SETX.*

We thank the reviewer for finding our work compelling and for their interest in the model. We have added significant body of data to consolidate the model as described below and detailed in the revised manuscript.

Major points:

1. *The direct physiological effect of USP11 on SETX ubiquitination is lacking. SETX expression is manipulated by ectopic overexpression so it is unclear whether USP11 directly acts on SETX. For example, when does USP11 associate with SETX (constitutive binding or only when RNA Pol I or Pol II is stalled, creating R-loops)? How does it associate or recognize SETX? What is USP11 doing to regulate SETX ubiquitination? If it isn't to deubiquitinate K48-linked SETX, then how is SETX degraded in a ubiquitin-dependent manner*

We thank the reviewer for these important points. Due to the large molecular weight of SETX and its low cellular abundance, biochemical studies have relied on ectopic expression^{1,2}. To examine association of endogenous SETX and USP11, as requested by the reviewer, we have employed proximity ligation assays (PLA)³. Under unperturbed conditions we detected a PLA signal between USP11 and SETX, indicating their cellular proximity (**Fig. 3c**). The induction of R-loops by CPT treatment led to ~5-fold increase in PLA signal. Furthermore, the SETX/USP11 interaction was transcription dependent because pre-treatment with the transcription inhibitor, α -amanitin, suppressed the PLA signal (**Fig. 3c**). These data indicate that endogenous SETX interacts with USP11 in a dynamic transcription-dependent manner that is promoted by R-loop induction.

To gain further insight into the mode of USP11/SETX interaction, we examined if it is mediated by the reported binding pocket for USP11⁴. The residues reported to mediate USP11 binding to its substrates were also required for binding to SETX, as shown by the lack of PLA signal between endogenous SETX and the USP11 binding site mutant, USP11 L208F/S242R (**Suppl. Fig. 4a**). Consistently, overexpression of wild type USP11, but not USP11 L208F/S242R, in USP11 knockout cells reduced R-loop accumulation (**Suppl. Fig. 4b**).

We have also significantly improved our detection protocol of K48-ubiquitin chains. We extended 10 μ M MG132 treatment from 2 to 16 hours and ran the input and pull-down lysates on separate SDS-PAGE gels rather than on a single gel. Keeping the inputs and pull-downs separately prevented masking the pull-down signal by the over-exposed inputs during chemiluminescent detection. As a result, we have now observed that loss of USP11 causes enrichment of K48-ubiquitinated SETX (**Fig. 4h, i**).

2. *Similarly, the link between KEAP1 and SETX (and USP11) is unconvincing. The transcriptional downregulation of Keap1 upon continuous passaging of USP11 KO cells should lead to decrease in endogenous KEAP1 proteins (this is not shown) (Figure 6F). The reduction of SETX upon HA-KEAP1 overexpression is also unconvincing (Figure 6F). How is loss of USP11 causing Keap1 transcriptional inhibition? Is R-loop formation at Keap1 gene preventing its normal expression? How is Keap1 promoting SETX ubiquitination? Does Keap1 associate with SETX constitutively or only after SETX is required to be downregulated when R-loop levels go down? Does SETX need to be dynamically ubiquitinated and regulated by an E3 ligase and DUB for R-loop homeostasis? What happens when Keap1 is downregulated in the presence or absence of USP11?*

Although KEAP1 was not the focus of this study, we have answered all reviewer questions, which significantly improved the manuscript. First, we indeed show that both KEAP1 mRNA and protein levels were reduced in aged USP11 KO cells (**Fig. 5e, f**). Second, we have repeated and quantified experiments with HA-KEAP1, which now convincingly show reduced SETX levels upon KEAP1 overexpression (**Fig. 5g**). Third, in **Fig. 7b-d**, we show through a series of DRIP and ChIP-qPCR experiments that USP11 depletion reduces SETX enrichment at *KEAP1* promoter. We further show that USP11 depletion increases R-loop accumulation, increases the double-strand break marker γ H2AX and the endonuclease XPF, at the *KEAP1* promoter. Together, these findings provide a plausible

explanation for the transcriptional downregulation of KEAP1 in USP11 deficient cells. Loss of USP11 not only increases SETX K48-ubiquitination and protein turnover, but also reduces the binding of the remaining SETX pool to *KEAP1* promoter, which leads to R-loop accumulation and subsequent processing by XPF to double-strand breaks, resulting in transcriptional downregulation.

We further show that KEAP1 overexpression increases K48-ubiquitinated SETX (**Fig. 6d**), which is consistent with data in **Fig. 5g** showing that over-expression of HA-KEAP1 reduces SETX protein levels, and with data in **Fig. 4h** showing that loss of USP11 increases K48-ubiquitination of SETX. Under unperturbed conditions SETX and KEAP1 were in close proximity as measured by PLA, and the interaction increased following R-loop induction by CPT (**Fig. 5h**). The interaction between KEAP1 and SETX is transcription dependent as the PLA signal was reduced by pre-incubation with the transcription inhibitor, α -amanitin (**Fig. 5h**). Finally, and as the reviewer requested, we examined what happens when KEAP1 is downregulated in the presence or absence of USP11. We depleted both separately and together (**Suppl. Fig. 5d**) and examined SETX ubiquitination using denaturing pull downs. Depletion of KEAP1 was sufficient to reduce the increased levels of SETX ubiquitination in USP11 deficient cells (**Fig. 6b**), indicating that KEAP1 and USP11 antagonistically regulate SETX ubiquitination. Consistently, depletion of KEAP1 led to a reduction of R-loop accumulation in USP11 deficient cells (**Fig. 6c**). Furthermore, overexpression of KEAP1 increased K48-ubiquitinated SETX, as measured by denaturing pull downs (**Fig. 6d**). Together, these data provide multiple lines of evidence demonstrating that KEAP1 opposes USP11 to regulate SETX ubiquitination and stability.

The increased global binding of SETX to both USP11 and KEAP1 following R-loop induction by CPT is intriguing. We suggest the extent of this binding is adjusted to favour more binding to USP11, thus increasing SETX levels, or more binding to KEAP1, thus reducing SETX level, at distinct genomic loci. This spatial regulation and control of SETX levels would ensure a fine balance to favour physiological R-loops that are required to promote transcription and at the same time suppress pathological R-loops that cause genomic instability and perturb transcription. A genome-wide DRIP and ChIP sequencing approach will address this idea that is born from our work and will be the subject of follow-up studies.

- 3. The functional readout for R-loop formation is lacking. It is unclear what an increase in R-loop intensity in nucleoli or at nuclear DNA regions is telling us about transcription or genome integrity. The authors use nucleoli foci number and intensity as their functional readout but R-loops themselves are nothing special. There is a natural dynamics in R-loop formation and dissociation at promoter and 3-end regions of transcriptionally active genes. Does disruption of USP11 or SETX or KEAP1 reveal ssDNA or dsDNA breaks at these R-loop sites, and lead to genomic instability? Or does USP11 or KEAP1 play a role in transcriptional termination and gene function at transcriptionally active gene sites with higher intensity R-loops?*

We thank the reviewer for the comment which we have now addressed by multiple experimental approaches. First, we performed alkaline comet assays to measure chromosomal breaks, which showed that the DNA damage in USP11 deficient cells can be suppressed by over-expression of RNase H1 (**Fig. 2h**). Second, we show that depletion of USP11 causes hypersensitivity to CPT and the PARP inhibitor, olaparib. The latter is consistent with a recent report⁵. Importantly, the hypersensitivity to CPT and olaparib was rescued by over-expression of RNase H1 (**Fig. 2i**), indicating that perturbed R-loop levels play a key role in the observed hypersensitivity. Third, we show through a series of DRIP and ChIP-qPCR experiments that USP11 depletion triggers accumulation of R-loops, the endonuclease XPF, and γ H2AX at the same genomic loci (**Fig. 7a-g**). XPF is a structure-specific endonuclease that has been reported to cleave R-loops to double-strand breaks⁶. The enrichment of XPF and γ H2AX was reduced by overexpression of RNase H1. This was true for all tested R-loop-positive but not negative loci, including two common fragile sites, *FRA16* and *FRA3B* (**Fig. 7g**). Thus, our data support a model in which loss of USP11 leads to R-loop accumulation, which are converted to double-strand breaks. Finally, we also show that depletion of SETX, USP11 or their co-depletion leads to similar reduced

levels of transcriptional read-through of the *ACTIN* gene (**Rebuttal Fig. 1**), indicating aberrant transcription rates due to R-loop accumulation. This is also consistent with the reduced transcription of *KEAP1* (**Fig. 5e**), the increased R-loops, DSBs and XPF at *KEAP1* promoter in USP11 deficient cells (**Fig. 7c-e**).

Rebuttal Fig. 1. USP11 depletion leads to reduced read-through of *ACTIN* gene. Control (Con si), USP11, SETX or USP11 and SETX-depleted MRC-5 cells were subjected to RT-qPCR using primers targeting exon 1 and 3' pause site of *Actin* gene. RT-qPCR values for the 3' pause were divided by values for exon 1, to measure full-length mRNA fragments (spanning exon 1 to the 3'-pause site), as described¹¹. $p > 0.05$, Student's t-test.

Reviewer 2

Jurga *et al* propose a **novel function** for the de-ubiquitylating enzyme USP11 in regulating senataxin (*SETX*) ubiquitylation, protein stability and, as a consequence, R loop levels. After monitoring the specificity of the R loop signal detected with the S9.6 antibody, the authors performed an siRNA screen to identify de-ubiquitylating enzyme that could affect R loop levels. They identify and further characterize USP11 function. In particular, they show that USP11 interacts with and de-ubiquitylates *SETX*, thus controlling R loop levels. They corroborate these results in CRISPR-knockout (KO) cells. However, cells adapt after several passages by reducing the levels of *KEAP1*, a putative E3-ligase for *SETX*.

1. Overall, the **amount of data generated is impressive**. The first set of data (R loop phenotype) is **quite convincing**, and it would be informative to further explore the well-established link between USP11 and the DNA damage. Is the DNA damage induction observed upon USP11 depletion (10.1074/jbc.M110.104745) mediated by R loop formation or by alternative mechanisms?

This is a great point, which we have now addressed. We show that USP11 deficient cells are sensitive to CPT and olaparib, which is rescued by the overexpression of the R-loop nuclease, RNase H1 (**Fig. 2i**), suggesting that USP11 depletion triggers DNA damage and cell death via R-loop accumulation. We complemented these experiments with alkaline comet assays and show that the DNA damage in USP11 deficient cells can be suppressed by over-expression of RNase H1 (**Fig. 2h**). Furthermore, we show that USP11 depletion increases R-loops, the endonuclease XPF, and γ H2AX at the same loci (**Fig. 7**). XPF is a structure-specific endonuclease that has been reported to cleave R-loops to double-strand breaks⁶. The enrichment of XPF and γ H2AX was reduced by overexpression of RNase H1. This was true for all tested R-loop-positive but not negative loci, including two common fragile sites, *FRA16* and *FRA3B* (**Fig. 7f, g**). Thus, our data support a model in which loss of USP11 triggers R-loops, which are converted to double-strand breaks.

2. The second part, in particular the impact of USP11 depletion on *SETX* turnover, is less convincing and needs additional controls. The ubiquitylation and interaction experiments have been mostly performed with the truncated *SETX*; what about the full-length?

We fully appreciate the reviewer comment and have now added significant body of new data to strengthen this part. We have improved, repeated and quantified the CHX-chase experiment in USP11 KO cells (**Fig. 3i, j**), which showed that the turnover of SETX, but not TDP1 as a control, was faster in CHX-treated USP11 KO cells. Furthermore, we significantly improved our detection protocol of K48-ubiquitin chains. We extended 10 μ M MG132 treatment from 2 to 16 hours and ran the input and pull-down lysates on separate SDS-PAGE gels rather than on a single gel. Keeping the inputs and pull-downs separately prevented masking the pull-down signal by the over-exposed inputs. As a result, we observed that loss of USP11 causes enrichment of K48-ubiquitinated SETX (**Fig. 4h, i**). Regarding full-length SETX, due to its large molecular weight and low cellular abundance, biochemical studies have relied on ectopic expression^{1,2}. However, to address the reviewer comment and examine association of endogenous full-length SETX and USP11, we have employed proximity ligation assays (PLA)³. Under unperturbed conditions we detected a PLA signal between USP11 and SETX, indicating their cellular proximity (**Fig. 3c**). Notably, the induction of R-loops by CPT treatment led to ~5-fold increase in PLA signal. Furthermore, the SETX/USP11 interaction was transcription dependent because pre-treatment with the transcription inhibitor, α -amanitin, suppressed the PLA signal (**Fig. 3c**).

3. *Due to the pleiotropic effect of the deubiquitylating enzymes, the final proof of the specificity of the observed phenotype would be rescue experiments in the USP11 KO with the WT or the USP11 defective for SETX binding.*

This would indeed be ideal. We initially relied on the widely accepted genetic epistasis to address this point, by showing that phenotypes caused by SETX, USP11 single mutants are similar to SETX/USP11 double mutants, which is depicted using multiple readouts (e.g., **Fig.3 and Fig.6**). Nevertheless, to provide a final proof as requested by the reviewer, we took advantage of the reported binding site mutant of USP11⁴. We first showed that wild-type USP11 can bind SETX whereas the USP11 binding site mutant, USP11 L208F/S242R, was not able to bind (**Suppl. Fig. 4a**). We then showed that over-expression of wild type USP11, but not USP11 L208F/S242R, was able to reduce the accumulation of R-loops in USP11 knockout cells (**Suppl. Fig. 4b**).

4. *KEAP1 downregulation as an adaptation mechanism in the USP11 KO cells is intriguing, although very preliminary*

Yes - fully agree, we were excited by these findings but digging into the mechanism was not really meant to be the focus of the study. However, we have now significantly strengthened this part. We show that both KEAP1 mRNA and protein levels were reduced in aged USP11 KO cells (**Fig. 5e, f**). We also repeated and quantified experiments with HA-KEAP1, which show reduced SETX levels upon KEAP1 overexpression (**Fig. 5g**). Furthermore, we show that USP11 depletion reduces SETX enrichment at *KEAP1* promoter and also increases R-loop accumulation, the double-strand break marker γ H2AX and the endonuclease XPF, at the same genomic sites (**Fig. 7b-d**). These observations support a model in which loss of USP11 not only increases SETX K48-ubiquitination and protein turnover, but also reduces the binding of the remaining SETX pool to *KEAP1* promoter, thereby increasing R-loop accumulation, which are then processed by XPF to double-strand breaks, resulting in adaptation by transcriptional downregulation.

5. *Additionally, the authors show that USP11 is enriched at genes with elevated R loop levels (figure 7a, not described in the results section). This is an interesting finding and could deserve further exploration (i.e. Does USP11 accumulation depend on SETX?)*

Thank you! This is a great point, which we have now described in the text and explored further, experimentally. USP11 was found to bind R-loop-positive loci such as *28S*, *R7*, *ING3*, *Actin*, *EGR1*, and also R-loop-negative loci such as *MYADM-neg*, *SNRPN-neg* (**Fig. 7a**). However, USP11 occupancy at R-loop-positive loci, but not R-loop-negative loci, was reduced following SETX depletion (**Fig. 7a**). This observation suggests that SETX promotes enrichment of USP11 at R-loops. To test whether SETX increases USP11 binding to chromatin globally, we performed nuclear fractionation assays. Depletion of SETX did not reduce USP11 levels in the insoluble chromatin fraction (**Supp. Fig. 6a**), suggesting that SETX regulates USP11 chromatin enrichment at specific genomic sites and not globally.

Next, we examined whether SETX binds to the same loci. As expected, SETX was enriched at R-loop-positive, but not negative, loci (**Supp. Fig. 6b**). We then examined whether enrichment of SETX on chromatin is USP11-dependent. Loss of USP11 reduced SETX binding to promoters (*KEAP1*, *EGR1*) and gene bodies (*ING3*) but not transcription end sites (*28S*, *R7*, *Actin*) (**Fig. 7b**). These findings not only provide a plausible explanation for the culture adaptation by downregulation of *KEAP1* transcription but are also exciting because SETX was previously shown to be recruited by BRCA1 to transcription termination sites to resolve R-loops⁷, however, how SETX is recruited to promoter-proximal regions and gene bodies remain unknown. Our ChIP-qPCR data identify USP11 as a putative candidate fulfilling this function by recruiting SETX to non-transcription end sites. A genome-wide approach will readily address this exciting idea, which is beyond the remit of the current study.

6. Finally, it could be relevant to test whether AOA2- and ALS4-associated mutations of SETX affect the interaction with USP11.

We managed to obtain ALS-4 patient fibroblasts that harbour the SETX L389S mutation, and matched controls⁸. We show that SETX binding to USP11 and to KEAP1 is increased in ALS4 patient-derived cells (**Rebuttal Fig. 2**). This is consistent with published immunoblotting data from ALS-4 fibroblasts, showing that the steady-state levels of ALS-4 SETX L389S protein were not different from matched controls⁸. SETX mutations that cause AOA-2 lead to very little, if any, SETX protein⁹ and therefore won't be relevant to this study.

Rebuttal Fig. 2. SETX interaction with USP11 and KEAP1 in ALS-4 patient fibroblasts. ALS-4 patient fibroblasts (ALS4) and matched-controls (Con) were subjected to proximity ligation assays using KEAP1, USP11, and SETX antibodies. Data are the average of 3 biological replicates \pm SD. ** $p < 0.01$, **** $p < 0.0001$; Student t-test.

Additional considerations:

1- Fig. 1a-b. It would be good to show CPT 30min images as well. The authors state that CPT treatment does not change the morphology of the nucleoli, a quantification would help.

The quantification has been provided for DMSO and CPT 10 min treatments (**Suppl. Fig. 1a**). Data related to CPT 30 min are included here as requested by the reviewer (**Rebuttal Fig. 3**) but removed from the manuscript because all subsequent experiments with CPT were performed at 10 min where we observed the maximum induction of R-loops.

Rebuttal Fig. 3. R-loop formation upon CPT treatment. MRC-5 cells were treated with 25 μ M CPT for 10 or 30 min and immediately harvested for S9.6 / nucleolin immunofluorescence. Representative confocal images are shown, scale bar is equal to 10 μ m.

2- *Supp. Fig. 3: The authors use the cycloheximide chase data to support the idea that SETX turnover is faster in USP11-depleted cells, however the treatments also impact on USP11 protein levels. It may be worth repeating this experiment in the early KO cells. Are the upper bands in the SETX blot modified forms? Can the author comment on the proteasome inhibition? USP11 depletion is barely observed in this condition.*

As suggested, we have repeated the experiment in KO cells (**Fig. 3i, j**), which shows that turnover of SETX is faster in USP11 KO cells compared to controls. We have highlighted using asterisks the non-specific bands and the bands that represent modified SETX throughout the paper (e.g. **Fig. 3d**). We have also commented on the proteasome inhibition and USP11 depletion in the text. The CHX-treatment in combination with 50nM USP11 siRNA was toxic. We therefore reduced the siRNA concentration to be able to perform this experiment. The CHX has worked as shown by the reduction of USP11 steady-state levels (**Suppl. Fig. 4e**) and the proteasome inhibitor worked as shown by the rescue of SETX accelerated turnover caused by CHX. We have now clarified these points in the text alongside USP11 expression.

3- *Fig. 3: Is the full-length SETX ubiquitylated as well?*

We now provide multiple lines of evidence indicating that full-length SETX is ubiquitylated. (1) Endogenous full-length SETX protein levels are reduced by USP11 depletion and restored by KEAP1 overexpression. (2) Endogenous full-length SETX protein turnover is accelerated by USP11 depletion. (3) Endogenous full-length SETX binds to USP11 and KEAP1 in a dynamic fashion that is dependent on transcription and promoted by R-loop induction. (4) Loss of USP11 reduced endogenous full-length

SETX binding to promoters (*KEAP1*, *EGR1*) and gene bodies (*ING3*) but not transcription end sites (*28S*, *R7*, *Actin*). This is exciting as it could potentially provide an explanation for how SETX is recruited to promoter-proximal regions and gene bodies, which is currently unknown.

4- *Suppl. Fig. 7a: showing in parallel wt control and non-complemented clones would be more convincing.*

The S9.6 slot blot (**prev. Suppl. Fig. 7a**, now **Suppl. Fig. 2d, e**) was used as a third orthogonal method to complement the IF and DRIP and confirm that USP11 KO cells complemented with active site mutant USP11 accumulate more R-loops than cells complemented with WT USP11. Repeating the S9.6 slot blot as suggested would show R-loop levels in control, USP11 KO and complemented USP11 KO cells, which have already been shown in **Fig 2e, f** (S9.6 immunofluorescence) and **Fig. 2g** (DRIP-qPCR).

5- *Fig. 5b. As above, it would be good to have a wt control and the non-complemented clones for a direct comparison of SETX expression*

Suppl. Fig. 4c now shows immunoblots from parental HEK-293, one USP11 KO clone and the corresponding complemented clones, wild type (WT) and USP11 active site mutant (USP11 C138S), which is consistent with **Fig. 3g**. Thus, both figures show that clones complemented with WT USP11 possess higher levels of SETX than clones complemented with USP11 active site mutant.

6- *Fig. 5d. As above. Moreover, inputs are missing.*

Apologies, we have added inputs to this figure, which is now **Fig. 4e, f**. We show the extent of ubiquitination of GFP-SETX in USP11-depleted cells and complemented USP11 KO cells (**Fig.4 a-f**). Please note that USP11 depletion in these experiments was very efficient (**Suppl. Fig. 3c**), acting as proxy for knockout.

7- *Fig. 6g. It would help the reader mentioning in the text why the RNF1681-110 and not the full-length is used.*

Absolutely, we have clarified this in the text. The truncated RNF168 construct expresses the catalytic domain of RNF168 (RING finger domain), which on its own is able to perform ubiquitination reactions but is less substrate-specific than the full-length RNF168¹⁰.

Reviewer 3

It is well established that R-loops occur as part of normal transcription participating in pausing of transcription and termination events to regulate the overall process. However, why they play beneficial roles their persistence can lead to genome instability so it is necessary to have suitable mechanisms in place to prevent their accumulation. This is mediated by enzymes such as senataxin and RNAse H which mediate their removal. The importance of such mechanisms is the occurrence of syndromes such as AOA2 and ALS4 associated with mutations in SETX the gene coding for senataxin. While the functioning of senataxin has received considerable attention over the last several years its regulation at the protein level has not been described.

The present submission addresses this issue in that it describes how the deubiquitinase (Dub) USP11 regulates senataxin proteostasis to control the accumulation of R-loops. The work described in the submission is **novel, very thorough and of interest** to those involved in studying the DNA damage response but also to a wider audience working on transcription regulation and consequences of deficiency in cell function and disease such as cancer. Use of statistics clear and convincing.

Specific comments

1. *This work provides new insight into how a specific Dub, USP11, for the helicase senataxin regulates the level of R-loops to minimize the risk of genome instability. It addresses this in a series of experiments by showing that deletion of USP11 results in ubiquitination and degradation of senataxin and as a consequence accumulation of R-loops.*

Thank you.

2. *The series of experiments designed to demonstrate this are well thought out and convincing.*

Thank you.

3. *In relation to Fig4a while the CRISPR –generated clones had no detectable USP11 the level of protein in the WT is also almost undetectable. This may be due to the high levels of overexpression of the catalytically inactive C318S mutant? It would be useful to have a blot with more convincing data for the WT. Also why is the mutant so highly expressed? How do you explain this by a positive-feedback loop?*

We have now provided more convincing images taken just after the complementation (**Fig. 2d**). As R-loop levels accumulate with prolonged USP11 loss, they lead to chromatin breaks (**Fig. 2h** and **Fig. 7d**). USP11 depletion has also been shown to cause defective homologous recombination (HR)⁵, suggesting that USP11 KO cells experience high-levels of stress, which is supported by our additional data showing induction of the senescence marker, p21 (**Suppl. Fig. 5c**). Therefore, it is plausible that USP11 KO cells upregulate active site mutant USP11 in a futile attempt to protect genome integrity via restoration of R-loop homeostasis and HR.

4. *In Fig 1 why the high background signal for S9.6 (R-loops) in the untreated control?*

We have provided a more convincing confocal image with less noise (**Fig. 1a**)

5. *There is some confusion in the positions of 250 markers and setx. For example in Fig 2f. There are two setx bands one ubiquitinated and the other not. Where should the 250 marker be? Below the main band?*

We apologise for the confusion. The markers have been properly aligned and non-specific bands denoted by asterisks throughout the paper.

6. *In Fig 6f we see two bands again. This time the 250 marker appears to coincide with the lower band. Is this setx or is it the upper band?*

Apologies again, the markers have been properly aligned and non-specific bands denoted by asterisks in this figure and throughout the paper.

7. *In further reference to Fig6f it is stated that this is “ Consistent with this model, overexpression of KEAP1 in aged USP11 knockout cells restored SETX protein to levels observed in young early passage cells (Fig. 6F). Restoration of setx is not obvious when considering either the upper or lower bands?*

We see the reviewer point and apologise for the confusion. We have changed this statement to ‘*KEAP1 overexpression in aged USP11 knockout cells **reduced**, rather than **restored**, SETX protein level*’.

8. *Nucleolar enrichment of R-loops already described (referred to in Fig 1) should be referenced e.g Yeo et al 2014.*

Yes indeed, really sorry for this oversight, the Yeo et al 2014 reference has been added.

9. *Can you comment further on cellular adaptation with respect in young versus old cell lines*

This is a very exciting finding which we have now explored further, experimentally. Through a series of DRIP-qPCR and ChIP-qPCR experiments we show that USP11 depletion increases R-loops, the endonuclease XPF, and γ H2AX at the same loci including the *KEAP1* promoter (**Fig. 7**). XPF is a structure-specific endonuclease that has been reported to cleave R-loops to double-strand breaks⁶. The enrichment of XPF and γ H2AX was reduced by overexpression of RNase H1. Thus, our data support a model in which loss of USP11 triggers R-loops at *KEAP1* promoter, which are converted to double-strand breaks, leading to transcriptional downregulation of *KEAP1*, thereby reducing *KEAP1* mRNA (**Fig. 5e**) and protein levels (**Fig. 5g**), in an attempt to counteract USP11 loss.

Once again, we thank the editors and reviewers for the valuable comments which have significantly improved the manuscript and we look forward to hearing from you.

Yours sincerely,

Sherif El-Khamisy

References

1. Yuce-Petronczki, O. & West, S. C. Senataxin, defective in the neurodegenerative disorder AOA-2, lies at the interface of transcription and the DNA damage response. *Molecular and cellular biology* **(2)**, 406–17 (2013).
2. Bennett, C. L. *et al.* Protein Interaction Analysis of Senataxin and the ALS4 L389S Mutant Yields Insights into Senataxin Post-Translational Modification and Uncovers Mutant-Specific Binding with a Brain Cytoplasmic RNA-Encoded Peptide. *PLoS One* **8**, e78837-10 (2013).
3. Abraham, K. J. *et al.* Nucleolar RNA polymerase II drives ribosome biogenesis. *Nature* 1–5 (2020) doi:10.1038/s41586-020-2497-0.
4. Spiliotopoulos, A. *et al.* Discovery of peptide ligands targeting a specific ubiquitin-like domain-binding site in the deubiquitinase USP11. *J Biol Chem* **294**, 424–436 (2018).
5. Orthwein, A. *et al.* A mechanism for the suppression of homologous recombination in G1 cells. *Nature* **528**, 422–426 (2015).
6. Cristini, A. *et al.* Dual Processing of R-Loops and Topoisomerase I Induces Transcription-Dependent DNA Double-Strand Breaks. *Cell Reports* **28**, 3167-3181.e6 (2019).
7. Hatchi, E. *et al.* BRCA1 Recruitment to Transcriptional Pause Sites Is Required for R-Loop-Driven DNA Damage Repair. *Molecular Cell* **57**, 636–647 (2015).
8. Grunseich, C. *et al.* Senataxin Mutation Reveals How R-Loops Promote Transcription by Blocking DNA Methylation at Gene Promoters. *Molecular Cell* **69**, 426-437.e7 (2018).
9. Suraweera, A. *et al.* Senataxin, defective in ataxia oculomotor apraxia type 2, is involved in the defense against oxidative DNA damage. *J Cell Biol* **177**, 969–979 (2007).
10. Horn, V. *et al.* Structural basis of specific H2A K13/K15 ubiquitination by RNF168. *Nat Commun* **10**, 1751 (2019).
11. Skourti-Stathaki, K., Proudfoot, N. J. & Gromak, N. Human Senataxin Resolves RNA/DNA Hybrids Formed at Transcriptional Pause Sites to Promote Xrn2-Dependent Termination. *Molecular Cell* **42**, 794–805 (2011).

REVIEWER COMMENTS

Reviewer #1 (Remarks to the Author):

The revised manuscript by Jurga et al has significantly improved from their initial submission. The conclusions are now strengthened by new experiments and results presented. The manuscript is now sufficient for publication.

Reviewer #2 (Remarks to the Author):

The revised version of the manuscript is improved and provides further insights into the function of USP11. In particular, the authors add that DNA damage induction observed upon USP11 depletion occurs via R loop accumulation (Fig. 2i, h and Fig. 7).

However, as already mentioned, the key experiment proving that USP11 de-ubiquitylates SENX (and KEAP1 ubiquitylates it) is an IP with the truncated SETX, what about the full-length? An IP of the full-length SENX is needed to address the point.

Additionally, it would be more convincing to see side by side the increase in SENX UB in the parental and KO cell lines and that this is rescued by WT USP11 and not by the catalytic mutant, as already suggested (Fig. 4e).

Detailed points:

- The chase experiment in USP11 KO cells is more convincing (Fig. 3i,j) than the KD experiment (Supp. Fig. 4d), which, as it is, is not conclusive since KD of the protein is not observed.
- Fig 1d, SENX blot is empty.
- Fig 2i. Can the authors comment on why they think RH1 overexpression in siCtrl cells increases sensitivity to olaparib?
- The authors conclude from the blot in fig. 4j that SENX interacts with endogenous USP11: it seems that USP11 has been probed after SENX since the SENX band is still visible in the blot, I wonder if the USP11 band in the IP is the upper SENX band. I would not be surprised that the interaction would be visible by PLA only, but I would be cautious in concluding interaction from these IP data.
- Why is actin IPed in Fig. 4a?
- Fig. 7a, IgG control for USP11 ChIP or a control showing an unrelated region with no USP11 CHIP signal is missing. Moreover, it seems the data are shown as fold change compared to 'Con si', therefore the y-axis is mislabelled.

General comments:

- Axis should always start from 0, otherwise they are misleading (i.e Fig. 4f/l, 5a/c/d, 6d, 7c...).
- For coherence, fig. 4h should show the bar + dots for each replica as well.
- Fig. 6b the bars are mislabelled
- Fig. 7b: Labelling of promoter/gene bodies in the main figure could be useful for the reader.

Reviewer #3 (Remarks to the Author):

The author has adequately addressed all the issues raised.

Re: Revision 2 – NCOMMS-20-26726

13 June 2021

Reviewer 1

The revised manuscript by Jurga et al has significantly improved from their initial submission. The conclusions are now strengthened by new experiments and results presented. **The manuscript is now sufficient for publication.**

Thank you!

Reviewer 3

The author has adequately addressed **all the issues raised.**

Thank you!

Reviewer 2

The revised version of the manuscript is improved and provides further insights into the function of USP11. In particular, the authors add that DNA damage induction observed upon USP11 depletion occurs via R loop accumulation (Fig. 2i, h and Fig. 7). However, as already mentioned, the key experiment proving that USP11 de-ubiquitylates SENX (and KEAP1 ubiquitylates it) is an IP with the truncated SETX, what about the full-length? An IP of the full-length SENX is needed to address the point. Additionally, it would be more convincing to see side by side the increase in SENX UB in the parental and KO cell lines and that this is rescued by WT USP11 and not by the catalytic mutant, as already suggested (Fig. 4e).

As the reviewer requested, we have now performed further experiments which demonstrate that transient USP11 depletion or KEAP1 over-expression increases ubiquitination of ectopically expressed **Full-Length Flag-SETX** (**Rebuttal Figure 1a, b**). We have included this additional data in supplementary figures (**Supp. Fig. 4d and 5d**). Furthermore, and to fully mitigate the reviewer's concern, we now also show that USP11 depletion or KEAP1 over-expression increases ubiquitination of **Full-Length endogenous SETX** (**Rebuttal Figure 1c, d**). We have included the new data in main Figures (**Fig. 4a and 6a**). The low SETX steady state level in SETX KO cells was insufficient to pull down and the ectopic overexpression of the 13 kb full-length Flag-SETX plasmid was too toxic to the CRISPR KO cells. Thus, we provide multiple lines of evidence to confirm that USP11 de-ubiquitylates senataxin; USP1 depletion increased the ubiquitylation of: (1) truncated N-terminal SETX 1-667, (2) full-length Flag-SETX, (3) Full-length endogenous SETX.

Rebuttal Fig. 1 USP11 and KEAP1 regulate the ubiquitination of full-length SETX. (a) Control (Con si) and USP11-depleted HEK-293 cells (USP11 si) were transfected with a plasmid encoding Ub-His and FLAG-SETX, lysed and subjected to nickel pull-down *under denaturing conditions* to purify ubiquitinated proteins. Blots were immunoblotted using anti-His and anti-Flag antibodies. The band intensities of Ubi-Flag-SETX were normalised to His-Ub and presented as fold increase of Flag-SETX ubiquitination compared to controls. Quantification was performed from 3 independent biological replicates and results presented as average +/- SD. (b) Same as in 'a' but cells were transfected with an empty vector (EV) or plasmids encoding HA-KEAP1. (c) Control (Con si) and USP11-depleted HEK-293 cells (USP11 si) were transfected with a plasmid encoding Ub-His, lysed and subjected to nickel pull-down *under denaturing conditions* to purify ubiquitinated proteins. Blots were immunoblotted using anti-His and anti-SETX antibodies. The band intensities of Ubi-SETX were normalised to His-Ub and presented as fold increase of SETX ubiquitination compared to controls. (d) Same as in 'c' but cells were transfected with an empty vector (EV) or plasmids encoding HA-KEAP1.

Detailed points:

- The chase experiment in USP11 KO cells is more convincing (Fig. 3i,j) than the KD experiment (Supp. Fig. 4d), which, as it is, is not conclusive since KD of the protein is not observed.

We agree with the reviewer and have therefore removed the KD experiment in Supp. Fig. 4d.

- Fig 1d, SENX blot is empty.

It is fine in our version, must be an issue during file conversion or upload – we trust it is now visible.

- Fig 2i. Can the authors comment on why they think RH1 overexpression in siCtrl cells increases sensitivity to olaparib?

Yes - we agree with the reviewer that this is an interesting observation. Over-expression of GFP-RNase H1 was toxic in siCtrl cells in which R-loops were not induced by USP11 KD or CPT. We suggest that the pool of overexpressed RNase H1 that is not engaged in dealing with an increased burden of R-loops, which is the case in siCtrl cells, becomes more available at Okazaki fragments, causing excessive processing, which generates cytotoxic PARP1 trapping intermediates during cell division that leads to the hypersensitivity (DOI: 10.1074/jbc.272.8.4647).

- The authors conclude from the blot in fig. 4j that SENX interacts with endogenous USP11: it seems that USP11 has been probed after SENX since the SENX band is still visible in the blot, I wonder if the USP11 band in the IP is the upper SENX band. I would not be surprised that the interaction would be visible by PLA only, but I would be cautious in concluding interaction from these IP data.

We have considered this possibility and repeated the experiment three more times. We probed the blots with USP11 antibodies first and then with GFP antibodies (for GFP-SETX1-667). The additional USP11 band was observed in all three biological repeats, in the IP lanes where GFP-SETX1-667 was over-expressed (Rebuttal Fig. 2). Although we agree with the reviewer that the PLA data is sufficient to show the interaction and we do not need an orthogonal IP confirmation, we feel confident to include the IP data based on the additional repeats.

Rebuttal Fig. 2 GFP-SETX¹⁻⁶⁶⁷ co-immunoprecipitates with USP11. GFP-empty vector (GFP-EV) or GFP-SETX¹⁻⁶⁶⁷- transfected HEK-293 cells were subjected to GFP immunoprecipitation. Three biological repeats are shown (Repeat 1, Rep2, Rep3). Membranes were probed with α -USP11 antibody first and then with α -GFP antibody.

- Why is actin IPed in Fig. 4a?

It is a remaining His-Ub signal as both antibodies are mouse, α -His was probed first.

- Fig. 7a, IgG control for USP11 ChIP or a control showing an unrelated region with no USP11 CHIP signal is missing. Moreover, it seems the data are shown as fold change compared to 'Con si', therefore the y-axis is mislabelled.

IgG control for USP11 ChIP has been added (Supp. Fig. 6a). Please note that USP11 was significantly bound over IgG in all tested loci. The y-axis label has been corrected.

General comments:

- Axis should always start from 0, otherwise they are misleading (i.e Fig. 4f/l, 5a/c/d, 6d, 7c...).

Both labelling is legitimately correct, but upon the reviewer's request, we have changed the figures so that the axis starts from 0.

- For coherence, fig. 4h should show the bar + dots for each replica as well.

Done

- Fig. 6b the bars are mislabelled

It has been corrected

- Fig. 7b: Labelling of promoter/gene bodies in the main figure could be useful for the reader.

We agree and have added the labelling as suggested.

Once again, we thank the editors and reviewers for the valuable comments which have significantly improved the manuscript and we look forward to further contributions to *Nature Communications*.

Yours sincerely,

Sherif El-Khamisy

REVIEWERS' COMMENTS

Reviewer #2 (Remarks to the Author):

The authors have now addressed all the issues raised.